

# Observation error estimation in climate proxies with data assimilation and innovation statistics

Atsushi Okazaki[1,2], Diego S. Carrió[3], Quentin Dalaiden[4], Jarrah Harrison-Lofthouse[5], Shunji Kotsuki[1,2], Kei Yoshimura[6]

[1]Institute of Advanced Academic Research, Chiba University, Chiba, Japan
[2]Center for Environmental Remote Sensing, Chiba University, Chiba, Japan
[3]Department of Physics, Universitat de les Illes Balears, Palma, Spain
[4]Nansen Environmental and Remote Sensing Center, Bergen, Norway
[5]School of Geography, Earth and Atmospheric Sciences, University of Melbourne, Melbourne, Australia
[6]Institute of Industrial Science, The University of Tokyo, Kashiwa, Japan

*Correspondence to*: Atsushi Okazaki (ats.okazaki@gmail.com)

**Abstract.** Data assimilation (DA) has been successfully applied in paleoclimate reconstruction. DA combines model simulations and climate proxies based on their error sizes. Therefore, the error information is crucial for DA to work optimally. However, little attention has peen paid to the observation errors in the previous studies, especially when the proxies are assimilated directly. This study assessed the feasibility of innovation statistics, a method developed for numerical weather prediction, for estimating observation errors in climate reconstruction and its impact on the reconstruction skills. For this purpose, we conducted offline-DA experiments over 1870-2000. Here, we assimilated stable water isotope records from ice cores, tree-ring cellulose, and corals. We found that the innovation statistics-based approach correctly estimated the observation errors, even with the offline-DA scheme. Although the accuracy of the estimation depended on the sample size and accuracy of the prior error covariance, the estimation generally improved the reconstruction skills. The reconstruction skills with the estimated observation errors were comparable to those with errors defined differently in the previous studies. In contrast with those methods used in previous studies, however, innovation statistics-based approach offers an objective and systematic way to estimate observation errors with light computational cost. As such, the innovation statistics-based approach should contribute to improving the reconstruction skills and observation networks.

## 1 Introduction

Data assimilation (DA) estimates the most likely states or parameters by combining prior information drawn from model simulations (background) and observations. DA is a well-established method in numerical weather prediction (NWP) (e.g., Houtekamer and Zhang, 2016; Kalnay, 2003; and references therein), and has recently been applied in, among other fields, paleoclimate reconstruction. Earlier studies have focused on the last millennium (e.g., Franke et al., 2017; Goosse et al., 2010,



2012; Hakim et al., 2016; Steiger et al., 2018; Tardif et al., 2019; Valler et al., 2022; Wu et al., 2025) while recent studies have started investigating deeper into the past (e.g., Annan et al., 2024; Kurahashi-Nakamura et al., 2017; Li et al., 2024; Mathiot et al., 2013; Osman et al., 2021; Renssen et al., 2015; Tierney et al., 2020; 2022).


The DA weighs the simulation and observations based on their errors in the estimation. Therefore, the error information is crucial for DA to work optimally. Let us introduce the definition of the errors in DA here to depict issues in paleo-DA. In DA, background error $\boldsymbol{\epsilon}^b$ and observation error $\boldsymbol{\epsilon}^o$ are defined as

$$\boldsymbol{\epsilon}^b \equiv \mathbf{x}^b - \mathbf{x}^t \tag{1}$$

$$\boldsymbol{\epsilon}^o \equiv \mathbf{y}^o - \mathcal{H}(\mathbf{x}^t) = [\mathbf{y}^o - \mathbf{y}^t] + [\mathbf{y}^t - \mathcal{H}^t(\mathbf{x}^t)] + [\mathcal{H}^t(\mathbf{x}^t) - \mathcal{H}(\mathbf{x}^t)] \tag{2}$$

where, $\mathbf{x}$ is the model state, $\mathbf{y}$ is the observations, and $\mathcal{H}$ is the observation operator that converts the model state to an observation-equivalent quantity. The superscripts $b$, $t$, and $o$ represent the background, the truth, and the observations, respectively. Observation error $\boldsymbol{\epsilon}^o$ consists of three distinct components: measurement error, which arises from instrumental limitations and observational noise; representativeness error, which reflects discrepancies between the model's spatial and
temporal resolution and the actual observations; and errors in the observation operator. Each component is represented by the term on the right side of Eq. 2 from left to right. The corresponding error covariance matrices denoted as $\mathbf{B}$ and $\mathbf{R}$ are defined as

$$\mathbf{B} = \langle \boldsymbol{\epsilon}^b (\boldsymbol{\epsilon}^b)^T \rangle \tag{3}$$

$$\mathbf{R} = \langle \boldsymbol{\epsilon}^o (\boldsymbol{\epsilon}^o)^T \rangle, \tag{4}$$

where the brackets $\langle \cdot \rangle$ denote a statistical expectation.

Here, we briefly review how $\mathbf{B}$ and $\mathbf{R}$ are treated in paleo-DA. Because of its common use, we focused on ensemble-based approaches in reviewing $\mathbf{B}$ (e.g., Franke et al., 2017; Hakim et al., 2016, Steiger et al., 2018; Tardif et al., 2019; Valler et al., 2022; 2024). Generally, background ensembles can be drawn from any collection of reasonable states. This may be a highly
informed prior, such as a short-term forecast from an accurate analysis as in NWP, or an "uninformed" prior, such as a random sample from a model climatology (Hakim et al., 2016). In paleo-DA, regardless of the accuracy of the model initial condition, the information will be lost long before the next analysis step because of the chaotic nature of the climate and temporarily sparse observations (typically, observations are available once a year). Therefore, it is meaningless to use the analysis as the initial condition for the subsequent model forecasting (Bhend et al., 2012). For this reason, "offline-DA" is commonly used,
where the background ensembles are drawn either from a single long climate simulation or from an ensemble of such simulations (e.g., Franke et al., 2017; Hakim et al., 2016, Steiger et al., 2018; Tardif et al., 2019; Valler et al., 2022; 2024) referred to as stationary offline-DA and transient offline-DA, respectively. Because there is no information other than external forcings (e.g., greenhouse gas concentrations, total solar irradiance, and orbital parameters) to constrain the model states, such



an "uninformed" background ensemble is suitable for representing the error or the uncertainty of the background. While several
studies have explored the feasibility of the online-DA and demonstrated its potential (e.g., Acevedo et al., 2017; Matsikaris et
al., 2015; Okazaki et al., 2021; Perkins and Hakim, 2017; 2021), offline-DA remains the preferred approach owing to its
simplicity and the low computational costs.

There is no established method for defining the observation error or observation error covariance matrix $\mathbf{R}$ in paleo-DA. The
most common method for specifying observation errors is to use residuals (e.g., Dalaiden et al., 2021; Franke et al., 2017;
Hakim et al., 2016; Osman et al., 2021; Steiger et al., 2018; Tardif et al., 2019; Perkins and Hakim, 2021; Tierney et al., 2020;
Valler et al., 2022). Here, the specified observation errors $\hat{\boldsymbol{\epsilon}}^o$ are given by the following form:

$$\hat{\boldsymbol{\epsilon}}^o = \mathbf{y}^o - \mathcal{H}(\mathbf{x}^{ref}) \tag{5}$$

where $\mathbf{x}^{ref}$ represents reference data. Many of the paleo-DA studies employ linear regression models as an observation
operators, where each proxy record is linearly regressed against reference data from the instrumental period (e.g., Dalaiden et
al., 2021; Franke et al., 2017; Hakim et al., 2016; Steiger et al., 2018; Tardif et al., 2019; Perkins and Hakim, 2021; Valler et
al., 2022). In this approach, observation-based gridded surface temperature data, such as HadCRUT5 (Morice et al., 2021), is
typically used as the reference data. Alternatively, process-based proxy system models (PSMs; Evans et al., 2013; Dee et al.,
2015) are sometimes used as observation operators (e.g., Acevedo et al., 2017; Dee et al., 2017; Okazaki and Yoshimura, 2017;
Steiger et al., 2017) to provide a more physically informed representation of proxy-climate relationships. A PSM provides a
complete set of forward and mechanistic processes by which climatic information is imprinted and subsequently observed in
proxy archives. Although they are more complex than the linear regression models, the same approach can be used to obtain
observation errors, provided that all the input variables for $\mathcal{H}$ are available at the proxy sites. However, it is rarely possible to
obtain all necessary variables directly from observations at proxy sites. In such cases, model simulations are instead used as
$\mathbf{x}^{ref}$ to provide the required inputs (Tierney et al., 2020; Osman et al., 2021).

Some studies have approximated observation errors by assuming that the representativeness error is dominant (e.g., Goosse et
al., 2012; Dalaiden et al., 2021; Rezsöhazy et al., 2022). These studies estimated representativeness errors at each observation
location by comparing two timeseries with different spatial representations, for instance, in-situ observation and the gridded
observation data or two simulations, one with high and the other with low spatial resolutions.

An alternative method for estimating observation error is based on the variance of the observation timeseries (Franke et al.,
2017; Okazaki and Yoshimura, 2017; Valler et al., 2022). In paleoclimate studies, it is common to show the observation noise
level as a function of the variance ($\sigma^2$) with SNR (signal-to-noise ratio) defined as $\sigma_{noise}/\sigma_{y^o}$, where the numerator and the
denominator are the standard deviation of the error and signal in a proxy timeseries, respectively (Smerdon et al., 2012 and
references therein). Typically, one-fourth of the variance, which corresponds to an SNR of 2, is assumed to be an observational





error in paleo-DA. The factor was decided based on measurement errors in Okazaki and Yoshimura (2017), while it was used without grounds for documentary-type observations in Franke et al. (2017) but verified later with the innovation statistics (Valler et al., 2022).


Although the aforementioned methods are widely accepted, they also have limitations. The first approach using linear regression models becomes impractical when the overlapping period between $\mathbf{x}^{ref}$ and $\mathbf{y}^o$ is too short. In general, climate proxies that span long periods tend to have low temporal resolution and few overlapping points, making it difficult to use this approach for deep climate reconstruction. Given that paleo-DA is also used to reconstruct deep-time paleoclimates, such as the Paleocene and Eocene (e.g., Li, et al., 2024; Tierney et al., 2022), different approaches are required. Besides, the observation error $\hat{\boldsymbol{\epsilon}}^o$ defined in Eq. 5 is different from $\boldsymbol{\epsilon}^o$ in Eq. 2, because $\mathbf{x}^{ref}$, whether it is based on observations or simulations, is not the truth. Because $\mathbf{x}^{ref}$ contains errors, the derived matrix $\mathbf{R}$ is likely to be overestimated. A few studies have introduced a scaling factor to compensate for overestimation, where a globally uniform factor is multiplied by all records to maximize the resultant analysis skill (Osman et al., 2021; Tierney et al., 2020). However, a globally constant scaling factor may not yield the best results. Although these factors may be individually tuned, manual tuning is unrealistically time-consuming. The second approach, which estimates observation errors based on the representativeness errors, requires a dense observation network, gridded observation datasets, or high-resolution model simulations, which are limited in terms of climate proxies or equivalent quantities. Additionally, the representativeness error may not always be dominant since its magnitude depends on the resolution of the model simulation used for the background, proxy type, and accuracy of the observation operators. The third approach, which assumes that the observation error is a fixed fraction of the total variance, is not well-supported by theoretical considerations, and there is no clear justification for its universal adoption. In addition, the manual tuning of this factor for each proxy record is impractical. Some studies use multiple error specification approaches, depending on the observation types (e.g., Dalaiden et al., 2021; Franke et al., 2017; Valler et al., 2022). However, hybrid approaches may introduce biases to specific types of observations, leading to suboptimal DA performance.


Wrongly specified observation error covariance matrix $\mathbf{R}$ can severely reduce paleoclimate reconstruction accuracy. Tierney et al. (2020) showed that the reconstruction skill score varied up to 20 % depending on the magnitude of error variance. We also confirmed that misspecified $\mathbf{R}$ can lead to skill score variations of up to 68 % (Fig. A1). The skill difference is as large as that between prior and analysis, highlighting the importance of accurate observation errors. Therefore, paleo-DA requires sophisticated and systematic methods to estimate observation errors.

In the field of NWP, several methods have been developed to estimate observation errors using innovation statistics (e.g., Dee and da Silva, 1999; Desroziers et al., 2005; Hollingsworth and Lönnberg, 1986). Here, the term "innovation" refers to the differences between the observations and background state in the observation space. Innovation statistics has been widely



adopted in many studies and weather forecasting centers (e.g., Honda et al., 2018; Lellouche et al., 2018; Minamide and Zhang, 2017; Okamoto et al., 2018; Schraff et al., 2016; Tandeo et al., 2020 and references therein).

This study investigated the feasibility of innovation statistics in estimating observation errors in paleo-DA and its impact on the reconstruction skill. For this purpose, we "reconstructed" climate for the 19[th] and 20[th] centuries, where abundant

instrumental data is available for verification. We adopted the ensemble-based offline-DA approach, in which isotopic proxies were assimilated using PSMs based on Okazaki and Yoshimura (2017).

The remainder of this paper is structured as follows: Section 2 describes the methods and the experimental design. Section 3 examines the accuracy of the observation error estimation and evaluates the reconstruction skills using a series of observing

system simulation experiments (OSSEs). Section 4 presents the estimation results obtained from the real observational data. Finally, Sections 5 and 6 present a discussion and summary of the study findings, respectively.

## 2 Methods

### 2.1 Local Ensemble Transform Kalman Filter (LETKF)

This study used the Local Ensemble Transform Kalman Filter (LETKF; Hunt et al., 2007), a variant of the ensemble Kalman

filter (EnKF), to solve the update equation of the Kalman filter. Let $\mathbf{x}$ be the $N$-dimensional model state and consider an ensemble of $m$-members. The analysis ensemble mean and the perturbations in the LETKF are given by:

$$\bar{\mathbf{x}}^a = \bar{\mathbf{x}}^b + \mathbf{X}^b \widetilde{\mathbf{P}}^a (\mathbf{H}\mathbf{X}^b)^T \mathbf{R}^{-1} \left( \mathbf{y}^o - \mathcal{H}(\bar{\mathbf{x}}^b) \right) \tag{6}$$

$$\mathbf{X}^a = \mathbf{X}^b \sqrt{m-1} \widetilde{\mathbf{P}}^{a 1/2}, \tag{7}$$

where, $\bar{\mathbf{x}}$ denotes the ensemble mean, and $\mathbf{X}$ denotes the ensemble perturbation matrix, and the superscripts $a$ and $b$ denote the

analysis and background, respectively. Here, $\bar{\mathbf{x}}$ is a vector of length $N$, and $\mathbf{X}$ is the $N \times m$ matrix whose $i$ th column is $\mathbf{x}^{(i)} - \bar{\mathbf{x}}$, where superscript $(i)$ denotes the $i$ th member of the ensemble ($i = \{1,2, \dots, m\}$). The notation $\mathbf{y}^o$ is the observation vector of length $p$, $\mathbf{R}$ the observation error covariance matrix whose size is $p \times p$, $\mathcal{H}$ the observation operator that converts the model state to the observation equivalent quantity, $\mathbf{H}$ the linearized observation operator whose size is $p \times N$, and $\widetilde{\mathbf{P}}^a$ the covariance matrix in the ensemble space whose size is $m \times m$ given by

$\widetilde{\mathbf{P}}^a = [(m-1)\mathbf{I}/\Delta + (\mathbf{H}\mathbf{X}^b)^T \mathbf{R}^{-1}(\mathbf{H}\mathbf{X}^b)]^{-1}, \tag{8}$

where, $\Delta$ is a covariance inflation parameter, which inflates the prior error covariance matrix to avoid underestimation of the analysis error covariance and filter divergence. To reduce the spurious error covariance among distant points due to sampling errors caused by the limited ensemble size, covariance localization has been commonly used in EnKF (e.g., Houtekamer and





Mitchell, 2001). Localization in the LETKF is implemented by inflating the observation error variance distant from the analysis

model grid point (Hunt et al., 2007; Miyoshi and Yamane, 2007). With localization, the observation error covariance matrix is replaced by

$$\mathbf{R} \leftarrow \boldsymbol{\rho}^{-1} \circ \mathbf{R}. \tag{9}$$

Here, $\boldsymbol{\rho}$ denotes the localization weights and is a function of the distance between the observations and the analysis model grid point. The LETKF solves the above equations at all model grid points by assimilating a subset of observations surrounding

each analysis grid point. Therefore, $\bar{\mathbf{x}}$ and $\mathbf{X}$ reduce to a scalar and $1 \times m$ matrix, in practice.

## 2.2 Innovation statistics

This study used the innovation statistics proposed by Desroziers et al. (2005) to estimate both observation errors and the covariance inflation factor. The observation-minus-background innovation vector ($\mathbf{d}_b^o$), given by the difference between the observations and the background state in the observation space, can be expressed as

$$\mathbf{d}_b^o \equiv \mathbf{y}^o - \mathcal{H}(\mathbf{x}^b) \cong \boldsymbol{\epsilon}^o - \mathbf{H}\boldsymbol{\epsilon}^b, \tag{10}$$

where $\boldsymbol{\epsilon}^o$ and $\boldsymbol{\epsilon}^b$ are observation and background errors, respectively and are defined as the difference from the truth (see also Eq. 1). Similarly, the observation-minus-analysis innovation vector ($\mathbf{d}_a^o$), the differences between observations and analysis in the observation space are given by

$$\mathbf{d}_a^o \equiv \mathbf{y}^o - \mathcal{H}(\mathbf{x}^a) = \mathbf{y}^o - \mathcal{H}(\mathbf{x}^b + \boldsymbol{\delta}\mathbf{x}^a) \cong \mathbf{y}^o - \mathcal{H}(\mathbf{x}^b) - \mathbf{H}\boldsymbol{\delta}\mathbf{x}^a = \mathbf{d}_b^o - \mathbf{H}\boldsymbol{\delta}\mathbf{x}^a. \tag{11}$$

The Taylor series expansion around $\mathbf{x}^b$ and Eq. 10 were used for the transformation. The notation $\boldsymbol{\delta}\mathbf{x}^a$ denotes the analysis increment and is defined as

$$\boldsymbol{\delta}\mathbf{x}^a \equiv \mathbf{x}^a - \mathbf{x}^b = \mathbf{K}[\mathbf{y}^o - \mathcal{H}(\mathbf{x}^b)] = \mathbf{K}\mathbf{d}_b^o, \tag{12}$$

where $\mathbf{K}$ is the Kalman gain given by

$$\mathbf{K} = \mathbf{B}\mathbf{H}^T(\mathbf{H}\mathbf{B}\mathbf{H}^T + \mathbf{R})^{-1}. \tag{13}$$

With the above equations, Eq. 11 can be further transformed into

$$\mathbf{d}_a^o \cong (\mathbf{I} - \mathbf{H}\mathbf{K})\mathbf{d}_b^o = \mathbf{R}(\mathbf{H}\mathbf{B}\mathbf{H}^T + \mathbf{R})^{-1}\mathbf{d}_b^o. \tag{14}$$

Finally, the differences between the analysis and background in the observation space ($\mathbf{d}_b^a$) can be derived using Eq. 14 as follows:

$$\mathbf{d}_b^a \equiv \mathbf{d}_b^o - \mathbf{d}_a^o \cong \mathbf{H}\mathbf{K}\mathbf{d}_b^o. \tag{15}$$





The innovation vectors defined in Eqs. 10, 14, and 15 can be used to derive several diagnostics based on the following
assumptions: (1) the observation and background errors are unbiased and uncorrelated, and (2) the observation and background
error covariances in the observation space ($\mathbf{HBH}^T$) are correctly specified. The first diagnostic considers $\mathbf{HBH}^T + \mathbf{R}$ and is
given as follows:

$$\langle \mathbf{d}_b^o (\mathbf{d}_b^o)^T \rangle \cong \langle \boldsymbol{\epsilon}^o (\boldsymbol{\epsilon}^o)^T \rangle + \langle \mathbf{H}\boldsymbol{\epsilon}^b (\mathbf{H}\boldsymbol{\epsilon}^b)^T \rangle = \mathbf{HBH}^T + \mathbf{R}. \tag{16}$$

Here, the cross-covariance terms are assumed to be zero because of the assumption (1). The background error covariance in
the observation space ($\mathbf{HBH}^T$) can be estimated using Eqs. 13, 15, and 16:

$$\langle \mathbf{d}_b^a (\mathbf{d}_b^o)^T \rangle \cong \mathbf{HK} \langle \mathbf{d}_b^o (\mathbf{d}_b^o)^T \rangle = \mathbf{HK}(\mathbf{HBH}^T + \mathbf{R}) = \mathbf{HBH}^T. \tag{17}$$

Finally, the observation error covariance ($\mathbf{R}$) can be estimated using Eqs. 14 and 16:

$$\langle \mathbf{d}_a^o (\mathbf{d}_b^o)^T \rangle \cong \mathbf{R}(\mathbf{HBH}^T + \mathbf{R})^{-1} \langle \mathbf{d}_b^o (\mathbf{d}_b^o)^T \rangle = \mathbf{R}. \tag{18}$$

Using the innovation statistics, the covariance inflation factor ($\Delta$) can be estimated adaptively (Li et al., 2009). This study
estimated the factor for each model grid point (i.e., locally) following Miyoshi (2011).

$$\Delta = \frac{trace\left(\langle \mathbf{d}_b^a (\mathbf{d}_b^o)^T \rangle \circ \boldsymbol{\rho} \circ \mathbf{R}^{-1}\right)}{trace\left(\mathbf{HBH}^T \circ \boldsymbol{\rho} \circ \mathbf{R}^{-1}\right)} \tag{19}$$

Here, ∘ denotes the Schur product. The inverse of $\mathbf{R}$ and $\boldsymbol{\rho}$ is multiplied for the normalization of multiple observations and
spatial smoothing of the inflation estimates. Note that $\mathbf{R}$ and $\mathbf{HBH}^T$ constitute the observations and simulated equivalent
quantities within the radius of influence.

## 2.3 Experimental design

We conducted two series of DA experiments, OSSE and REAL, which only differ in the observations to be assimilated. In
OSSE, we created the observations using a reference simulation (so-called "nature run") and observation operators. The
background ensemble was created using the same model as that for the nature run (i.e., perfect model experiment), but with
slightly different external forcings (see below for more details). This allows for a direct evaluation of the observation error
estimation in the absence of other error sources. We also performed climate reconstruction with the estimated observation
errors in REAL, which uses real observation data for DA. In addition to the differences in observations, the OSSE and REAL
experiments shared the same experimental settings. Multiple experiments were conducted for each framework. The default
experimental settings and experiment-specific configurations are detailed in Sections. 2.3.1 and 2.3.2, respectively.

### 2.3.1 Default experimental settings

**Background ensemble**: We constructed a background ensemble using MIROC5-iso (Okazaki and Yoshimura, 2017; 2019),
an isotope-enabled atmospheric general circulation model (GCM) developed based on the atmospheric component of MIROC5



(Watanabe et al., 2012). MIROC5-iso is forced by simulated SST and sea ice concentrations (SIC), observed greenhouse gases (carbon dioxide, methane, and chlorofluorocarbons), ozone, and land-use changes. We derived SST and SIC data from
historical simulations in the Coupled Model Intercomparison Project Phase 5 (CMIP5; Taylor et al., 2012). Specifically, we used the MIROC5 simulation with the "r1i1p1" ensemble member. The isotopic compositions of sea surface water and sea ice were kept constant and assumed to be 0‰ and 3‰, respectively (Joussaume and Jouzel, 1993). The model resolution was set to T42 (~280 km on the equator), with 40 vertical levels. We used a single-member simulation covering 1870-2005 to generate a 136-member background ensemble, where each year of the simulation served as an ensemble member (Steiger et al., 2014).
We used the same background ensemble for all experiments, unless otherwise noted.

**Observations**: We used the Iso2k (Konecky et al., 2020) and PAGES2k databases (PAGES2k Consortium, 2017) for the observation data. The Iso2k database contains stable oxygen ($^{18}$O) and hydrogen ($^{2}$H) isotopic records from various archives. This study used isotopic records of ice cores, corals, and tree-ring cellulose, as described by Okazaki and Yoshimura (2017; 2019). From the PAGES2k database, only three documentary surface temperature records are used, complementary. In the
REAL experiments, observations with a temporal resolution shorter than 1 year were averaged to obtain annual means, whereas observations with a resolution longer than 1 year were discarded. In OSSEs, observations were made based on the nature run, which was constructed using a simulation with a configuration identical to that for the background except for SST and SIC; the MIROC5 simulation of the CMIP5 historical run with the ensemble member "r2i1p1" was used. The model fields at the proxy locations were extracted using simple linear interpolation and then converted to an observation-equivalent quantity using
observation operators, or PSMs, as described below. The input and output variables for the PSMs were monthly means and were annualized such that the experimental setting was comparable to that for REAL. To simulate the observational uncertainty, random noise was added to each data. This noise was drawn from a normal distribution with zero mean and variance equal to one-fourth of the temporal variance of the annualized time series. The number of observations and the spatial distribution are shown in Fig. 1.

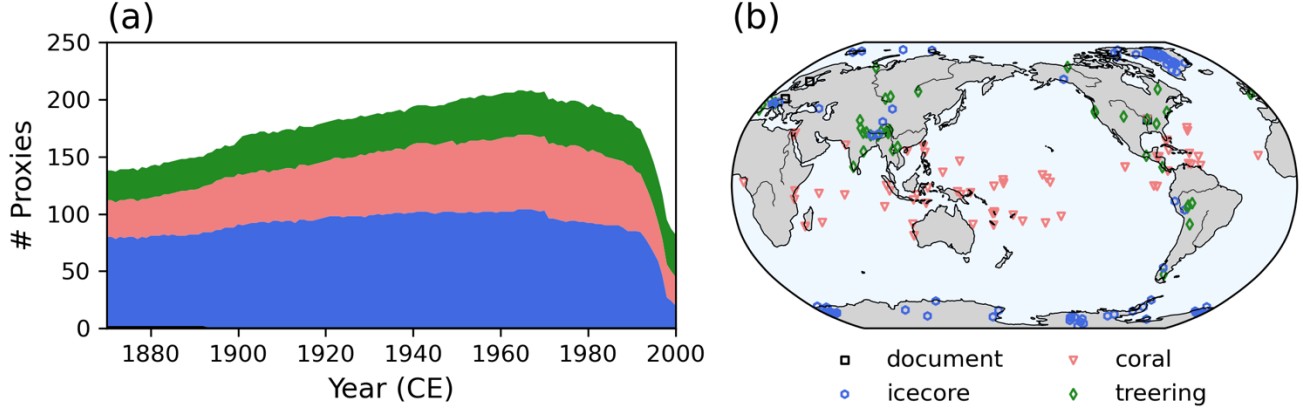


**Figure 1: (a) Number of proxies used in this study. The color shows the type of proxy. (b) Geographical location of proxies.**



**Observation operators (PSMs)**: We used the PSM developed by Liu et al. (2013, 2014) for corals, and that of Roden et al.
(2000) for tree-ring cellulose. For the ice cores, we assumed that the isotopic composition was the same as that of precipitation at the time of deposition. In reality, the isotope ratios in ice cores may deviate from those in precipitation due to post-depositional processes (e.g., Schotterer et al., 2004). More detailed information on PSMs can be found in Okazaki and Yoshimura (2017; 2019). For surface temperature, simulated 2-m temperature is directly used.

**Data assimilation**: The assimilation was conducted following the anomaly-DA approach (e.g., Keenlyside et al., 2008; Smith
et al., 2007), where the corresponding climatological mean is subtracted from both observations and background to mitigate the detrimental impact of model bias. The climatological mean was computed using the overlapping years between each observation and the simulation for 1900-2000. For the background covariance localization, a fifth-order polynomial function was used (Gaspari and Cohn, 1998). The localization scale was manually tuned beforehand, and a half-localization scale of 8,000 km was used for all the experiments. The observation error covariance matrix was assumed to be diagonal, as in many
other studies (e.g., Franke et al., 2017; Hakim et al., 2016, Steiger et al., 2018; Tardif et al., 2019; Valler et al., 2022). The diagonal element of the matrix (i.e., error variances) was set to one-fourth of the temporal variance of the annual observation timeseries, unless otherwise specified. This size of the error variance was identical to that used to create the observations in the OSSE. The assimilation was conducted for 1870-2000 in both OSSE and REAL experiments.

### 2.3.2 List of experiments

In the OSSE, we performed the "EST" and "BIAS" experiments. In EST, the observation errors were estimated using innovation statistics and the performance of the climate reconstruction with DA with estimated observation errors was assessed. In BIAS, we investigated the impact of a biased background error covariance. In the REAL, observation errors defined in different ways as in previous studies (UNI, VAR, and RES) were tested and compared with "EST". Each experimental setting is detailed below and summarized in Table. 1.

**EST**: We estimated observation errors based on innovation statistics (Eq. 18). To address sampling error in the estimation, a large number of samples is required for reliable estimates. We used the entire period of one DA experiment (i.e., 1870-2000; 131 samples) to maximize the sample size. We also estimated the covariance inflation factor as well (Li et al., 2009) since the simultaneous estimation of the covariance inflation factor and $\mathbf{R}$ improves the analysis skills. The set of DA experiment and the estimation of the inflation factor and $\mathbf{R}$ can be conducted iteratively, where the estimated inflation factor and $\mathbf{R}$ from one
iteration are used in the subsequent DA experiment. In this study, we repeated the procedure 10 times. In the OSSE, $\mathbf{R}$ used in the first iteration ($\mathbf{R}_{ini}$) of the DA is given by either of 4-times of variance (Rx16), one-fourth the variance (Rx1), or one-sixteenth the variance (Rx0.25). In Rx1, $\mathbf{R}_{ini}$ is equal to the actual error ($\mathbf{R}_{tru}$). No covariance inflation was applied in the first iteration for either in the OSSE or REAL.




**BIAS**: We conducted an experiment similar to EST but with a biased background ensemble to examine the impact of the
biased off-diagonal term of **B**. Instead of the MIROC5-iso simulation used in EST, the model simulation forced by observation-
based SST and SIC from HadISST1 (Rayner et al., 2003) was used. The experiment was conducted only for the OSSE.

**UNI**: In this experiment, each observation type shared the same observation errors, given by the mean of the estimated values
from EST multiplied by the globally constant scaling factor $k$ (0.25, 0.5, 0.6, 0.7, 0.8, 0.9, 1, 1.5, and 2). The experiment was
conducted only for the REAL.

**VAR**: Observation errors were determined based on the standard deviations of each proxy record multiplied by a globally
constant scaling factor (Franke et al., 2017; Okazaki and Yoshimura, 2017; Valler et al., 2022). The scaling factor $k$ was varied
at 1/8, 1/4, 1/2, 3/4, 1, 2, 4, 8, and 16. The experiment was conducted only for the REAL.

**RES:** In this experiment, observation variances are given as follows:

$$\mathbf{R} = k \, diag(\langle \boldsymbol{\epsilon}\boldsymbol{\epsilon}^T\rangle), where \, \boldsymbol{\epsilon} = \boldsymbol{y}^O - \mathcal{H}(\mathbf{x}^{ref}). \tag{20}$$

We used a simulation forced by observation-based SST and SIC, HadISST1 (Rayner et al., 2003), as $\mathbf{x}^{ref}$ instead of gridded
instrumental observation data. The other simulation settings were identical to those for the background ensemble. Only the
records that overlapped with the simulation for at least 30 years were used. We used this approach because no gridded isotopic
observation dataset is available to drive the observation operators. The scaling factor $k$ was varied at 0.25, 0.5, 0.6, 0.7, 0.8,
0.9, 1, 1.5, and 2. The experiment was conducted only for the REAL.

**Table 1: Experimental settings**

| Exp. Type | Exp. Name | Background | Observation | R |
|---|---|---|---|---|
| OSSE | EST | r1i1p1[a] | r2i1p1[b] | Estimated with innovation statistics |
| | BIAS | HadISST1[c] | r2i1p1[b] | Estimated with innovation statistics |
| REAL | EST | r1i1p1[a] | Iso2k[d] | Estimated with innovation statistics |
| | UNI | r1i1p1[a] | Iso2k[d] | Same observation errors for each observation type |
| | VAR | r1i1p1[a] | Iso2k[d] | Variance of the proxy timeseries multiplied by a scaling factor |
| | RES | r1i1p1[a] | Iso2k[d] | Based on the difference between observation and a reference simulation |

[a]*MIROC5-iso forced by a historical run of MIROC5 in CMIP5, labelled "r1i1p1"*

[b]*Iso2k-like observation created with MIROC5-iso forced by a historical run of MIROC5 in CMIP5, labelled "r2i1p1"*

[c]*MIROC5-iso forced by observation-based SST and SIC of HadISST1 (Rayner et al., 2003)*

[d]*Konecky et al. (2020)*



## 2.4 Metrics

We verified the results for the annual mean 2-m air temperature against the reference data $\mathbf{x}^{ref}$. Three metrics were used to evaluate the reconstruction skills; the Pearson correlation coefficient (*CC*), the coefficient of efficiency (CE; Nash and Sutcliffe, 1970), and the relative variance (RV). The definitions are as follows:

$$CC = \frac{\sum x_i^a x_i^{ref}}{\sum (x_i^a)^2 \sum (x_i^{ref})^2},$$

$$CE = 1 - \frac{\sum (x_i^a - x_i^{ref})^2}{\sum (x_i^{ref})^2}, \text{ and}$$

$$RV = \frac{\sum (x_i^a - \bar{x}^a)^2}{\sum (x_i^{ref} - \bar{x}^{ref})^2}.$$

Here, $x_i^a$ and $x_i^{ref}$ denote $i$ th model grid point of the analysis and the reference. In all the metrics, the inputs are given in anomaly forms with respect to the climatological mean to ignore the model biases. For the OSSE experiments, the nature run was used as the reference data. For the REAL, HadCRUTv5 (Morice et al., 2021) was used as the reference data.

## 3 Perfect model results

This section evaluated the performance of the innovation statistics in estimating the observation errors and its effects on the reconstruction skills in the OSSE experiments.

Figure 2a compares the true observation variances ($\mathbf{R}_{tru}$) and the estimated observation variances ($\mathbf{R}_{est}$) for EST with an initial $\mathbf{R}$ 16-times larger than $\mathbf{R}_{tru}$ (Rx16). Most of the estimated observations fall between the two reference lines in Fig. 2a with the slopes of 2 and 4. Given that the initial observation variances were 16-times larger than the truth, the differences between $\mathbf{R}_{est}$ and $\mathbf{R}_{tru}$ became smaller than $\mathbf{R}_{ini}$ after the estimation. Improvements were observed across different proxy types, demonstrating that the estimation method functioned properly.





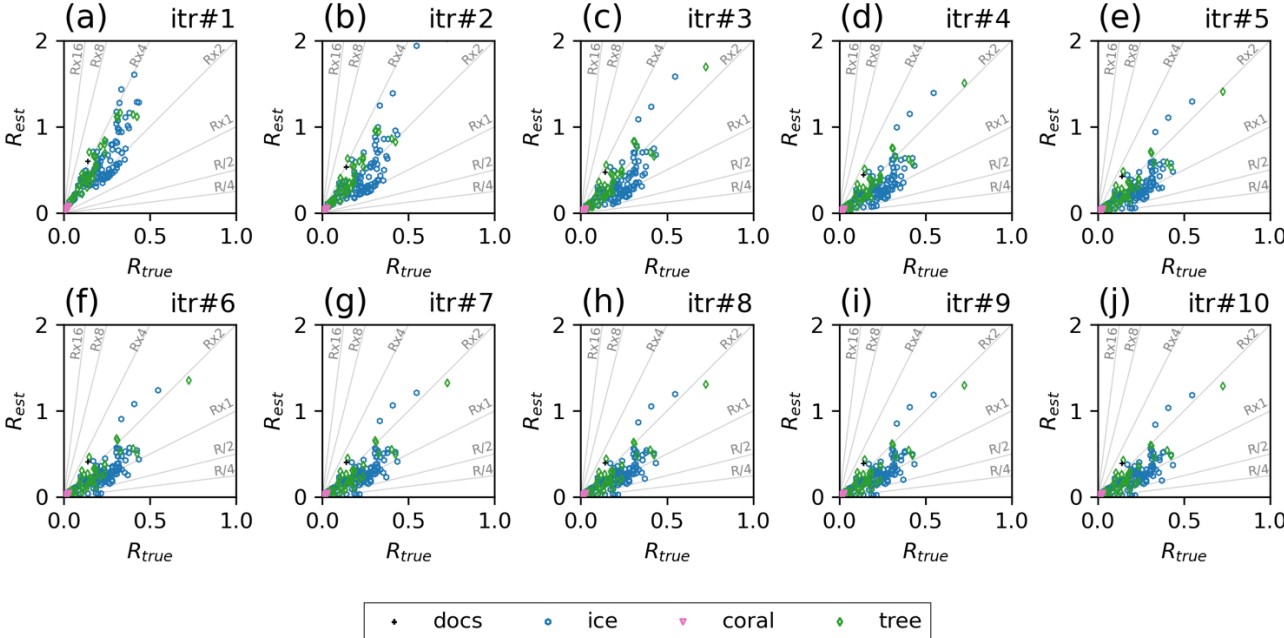

**Figure 2: Relationship between true and estimated observation error variances at each iteration step for the OSSE with initial**
**observation variance of Rx16.**

With a more accurate and representative **R**, the DA is expected to become more accurate. Moreover, the observation error

estimates are expected to be more accurate when **R** is correctly specified (Desroziers et al., 2005). Accordingly, we conducted

a similar DA experiment but with $\mathbf{R}_{est}$ instead of $\mathbf{R}_{ini}$. We also applied the estimated prior error covariance inflation factors in

the DA, following Li et al. (2009). After analyzing the 131-year states with $\mathbf{R}_{est}$ and inflation factors, we estimated both **R** and

the inflation factors again using a new set of $\mathbf{x}^{a}$ and $\mathbf{x}^{b}$. We repeated the procedure 10 times. Iteratively applying the innovation

statistics further improved the accuracy of the observation error estimates (Figs. 2b-j and 3). After the 5[th] and 6[th] iterations, the

estimated errors converged. The ratio of $\mathbf{R}_{est}$ to the $\mathbf{R}_{tru}$ was 0.1-3.5 after the 10[th] iteration, with a mean absolute percentage

error of ~46%. The remaining inaccuracies can be attributed to sampling errors associated with the limited sample size. A

more detailed discussion of this limitation is provided in Sect. 6.2.




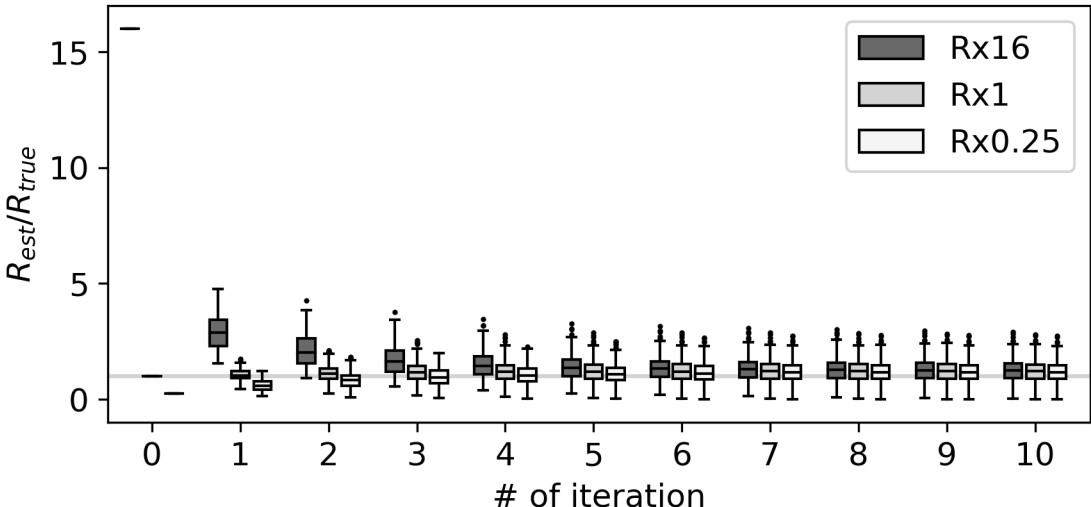

**Figure 3: Ratio of the estimated observation error variance according to iteration in OSSE Rx16 (dark grey), Rx1 (grey), and Rx0.25 (white). Horizontal bars of each box correspond to the 1$^{st}$, 50$^{th}$, and 99$^{th}$ percentiles. Data smaller (larger) than the 1$^{st}$ (99$^{th}$) percentile are plotted as dots. Grey horizontal line shows a ratio of 1. The values shown at 0$^{th}$ iteration are R$_{ini}$/R$_{tru}$.**

The reconstruction skills improved for all metrics with $\mathbf{R}_{est}$ (Fig. 4). With iteration, the global mean CC increased from 0.55 to 0.58, and CE improved from 0.28 to 0.37. The best skill scores were obtained when $\mathbf{R}_{est}$ was estimated using $\mathbf{x}^a$ and $\mathbf{x}^b$ from the 5$^{th}$ iteration. After peaking, the skill scores gradually decreased with further iterations. This could be due to the overfitting of $\mathbf{R}_{est}$ associated with the sampling bias. The largest improvements were observed in the tropics, particularly in the western tropical Pacific Ocean (Fig. 5). This is likely due to the significant impact of corals on reconstruction skills (Okazaki and 340 Yoshimura, 2017; Shoji et al., 2022).



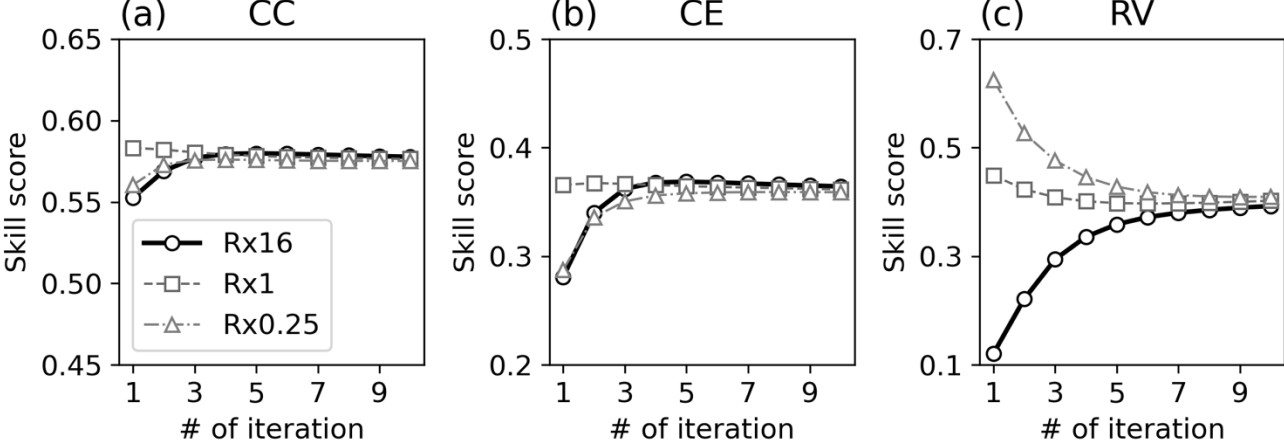

**Figure 4: Reconstruction skill score of (a) CC, (b) CE, and (c) RV according to iteration in OSSE Rx16 (thick black line with circle), Rx1 (thin dashed grey line with square), and Rx0.25 (thin dashed-dotted grey line with triangle).**

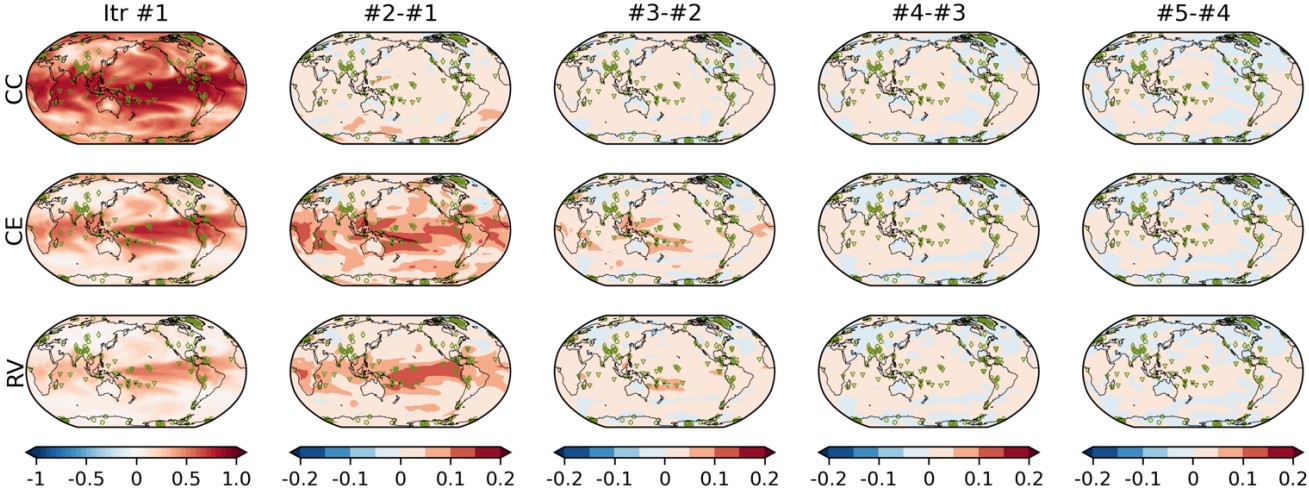

**Figure 5: Reconstruction skill score of (top) CC, (middle) CE, and (bottom) RV in OSSE EST Rx16. Left column shows the reconstruction skill at the 1st iteration (i.e., without observation error estimation). Second to fifth columns show the skill difference from the previous iteration; for instance, the second column shows the difference between 2nd and 1st iterations. Markers in each map show the position of proxies.**





To assess the dependence on the initial observation error covariance, we repeated the same test using two different $\mathbf{R}_{ini}$. The results showed a similar tendency in Rx0.25 and Rx16, where both observation error estimates and the reconstruction skills improved with iterations (Figs. 3 and 4). In contrast, $\mathbf{R}_{est}$ gradually deviated from the $\mathbf{R}_{tru}$ with increasing iterations in Rx1, where $\mathbf{R}_{ini} = \mathbf{R}_{tru}$, leading to a slight decrease in the reconstruction skill scores. This is due to the sampling error reducing estimation accuracy, as shown in Rx16. A notable difference in RV was found among the experiments, which increased with

iterations in Rx16, whereas it decreased in the others. This can be explained by how the DA weighs the model prior and observations based on their errors: Larger observation errors result in lower observation weights. In Rx16, the observation error variances in the analysis step of the 1st iteration are intentionally overestimated, resulting in the small difference between the analysis and the prior. Because the prior was the same at all the analysis steps, the analysis also remains nearly stationary, leading to a low RV. With iteration, the observation error decreased, allowing the DA to assign more weight to the observations,

in turn increasing the RV. The opposite occurred in Rx0.25, where the observation errors were initially underestimated. With increasing iterations, the estimated observation errors increased, shifting more weight onto the prior in the DA process. Consequently, the RV decreased with iterations in Rx0.25. Despite these differences, all the experiments ultimately converged to the same $\mathbf{R}_{est}$ and the skill scores, suggesting the little dependency on $\mathbf{R}_{ini}$ including RV.

## 4 Real observation results

Observation errors were estimated using innovation statistics, as in Sect. 3, but for real observations. The estimated errors after the 10th iteration ranged from 0.79 ℃ to 2.11 ℃ (mean of 1.59 ℃) for surface temperature, from 0.45 ‰ to 3.50 ‰ (mean of 1.42 ‰) for ice cores, 0.02 ‰ to 0.44 ‰ (mean of 0.12 ‰) for corals, and 0.07 ‰ to 1.54 ‰ (mean of 0.82 ‰) for tree-ring cellulose (Fig. 6a and 6b). Errors can be expressed as the ratio of $\mathbf{R}_{est}$ to the variance of each observation. At most locations, the estimated observation errors exceeded one-fourth of the variance, which is the typical value for paleo-DA (Figs. 7c and

7d). Although the estimated observation errors were system specific, as shown in Eq. 2, these results suggest a need for reconsidering the size of observation errors in paleo-DA. The corresponding SNR were approximately 1.25, and 1.22 for the corals and tree-ring cellulose, respectively, whereas it was approximately 1.0 for ice cores, indicating that either measurement, representativeness, or PSM errors for ice cores is larger than those for the other proxy types. This is consistent with previous results showing that the reproducibility of ice cores is lower than that of other proxies (Okazaki and Yoshimura, 2019). The

estimated SNR for the surface temperature (1.14) was smaller than that estimated by Valler et al. (2022) (2), indicating that the estimated error was greater in the current study. This was likely due to the coarser horizontal resolution of the background in this study (T42) compared with that in Valler et al. (2022) (T63).





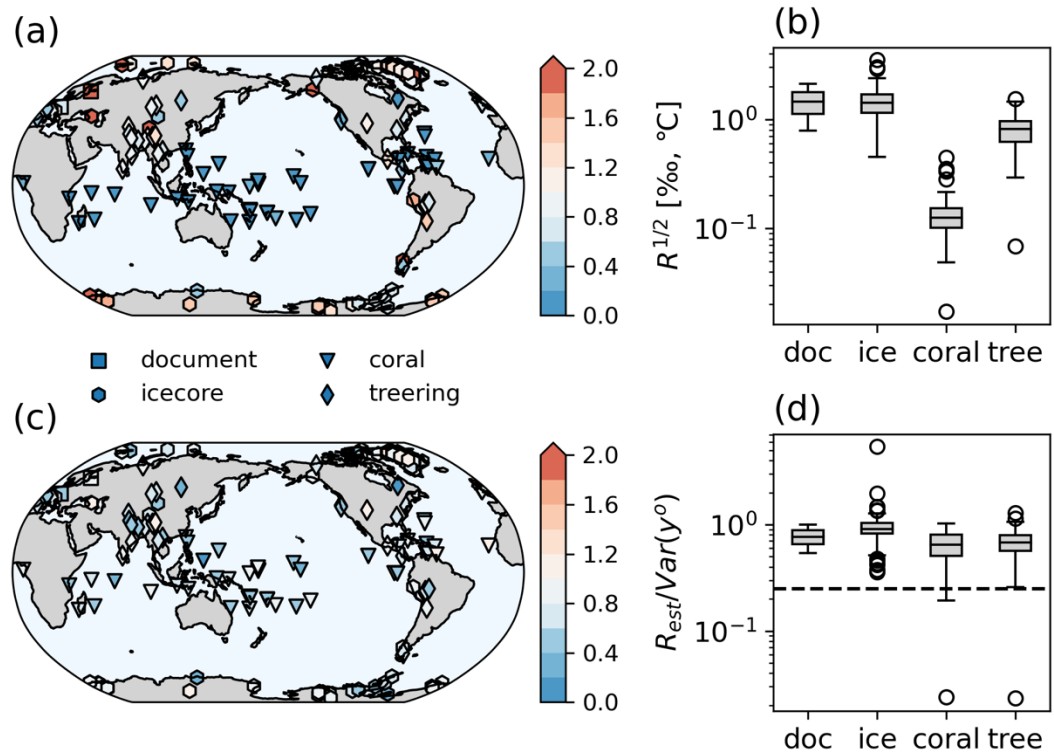

**Figure 6: Estimated observation error at the 10th iteration in the REAL experiment as the (a, b) ratio to the temporal variance of each observation and (c, d) physical unit (‰). Itr=10.**

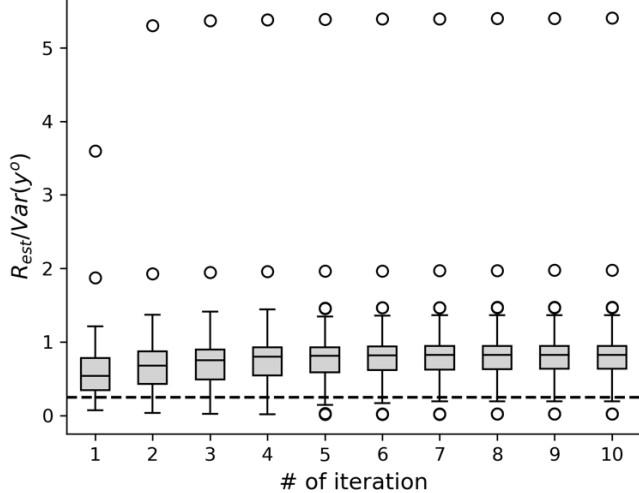

**Figure 7: Similar to Fig. 3 but for REAL experiment. Estimated observation error variances are normalized to temporal variance of corresponding proxy. Horizontal dashed line shows the one-fourth of the variance, corresponding to SNR = 0.5.**




The reconstruction skills for surface temperature are shown as a function of the number of iterations in Fig. 8. Here, the scores were computed against HadCRUT5 (Morice et al., 2021) for 1960-2000. This period was selected for better spatial coverage of the reference data. The skill scores increase with each iteration, regardless of the metrics used. Without error estimation, the CC and the CE scores were 0.38 and 0.04, respectively. After iteration, these values increased to 0.43 and 0.16, respectively.

The skill scores were relatively low compared to those of previous paleo-DA studies (e.g., Steiger et al., 2018; Tardif et al., 2019; Valler et al., 2024), likely due to the limited number of assimilated observations. The most notable improvements occurred in the tropical Atlantic Ocean, North Africa, and the north-eastern part of North America, where the initial CC and CE scores were relatively low (Fig., 9). In these regions, the estimated observation errors increase with iterations, effectively reducing the detrimental increments in the DA process and improving the reconstruction skill.


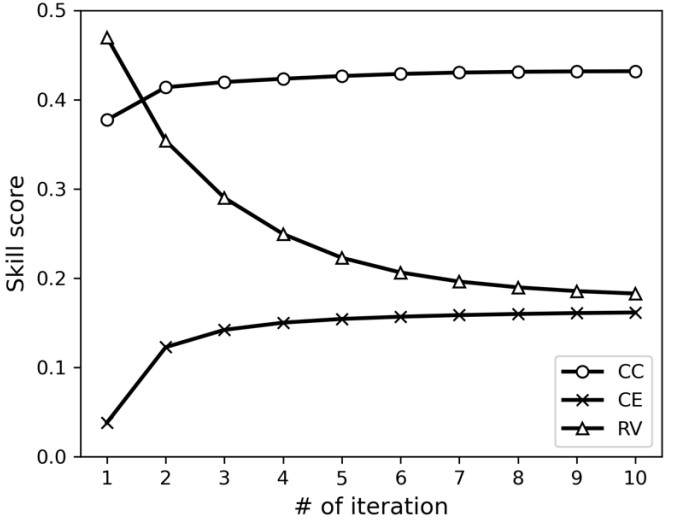

**Figure 8: Reconstruction skill score of CC (circle), CE (triangle), and RV (square) in the REAL experiment according to iteration.**



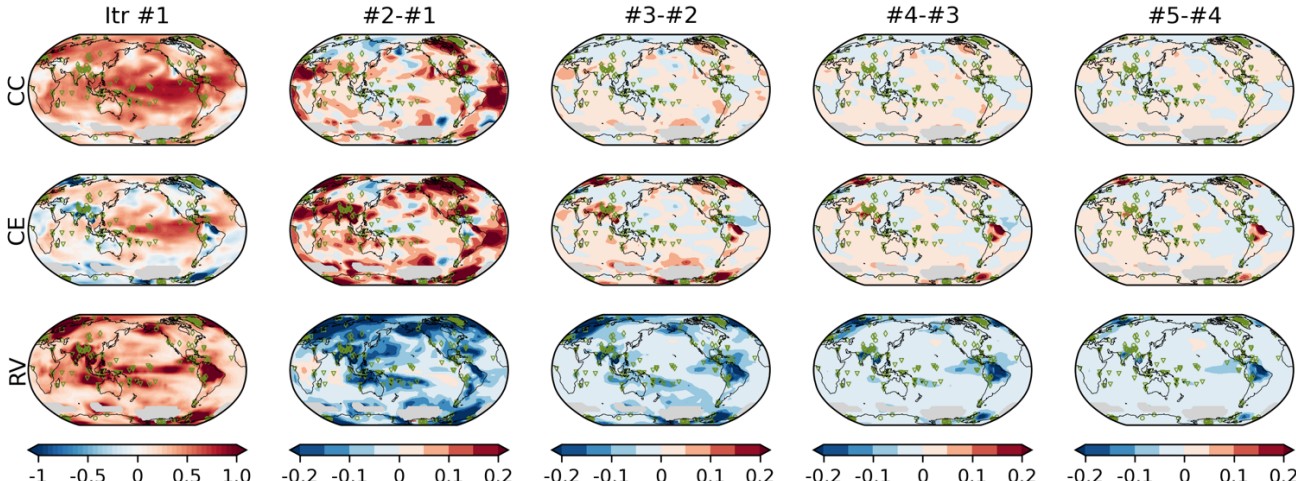

**Figure 9: Similar to Fig. 5 but for the REAL EST experiment. The scores were calculated against HadCRUTv5 (Morice et al., 2021) for 1960-2000. Areas where observations covers less than half of the period are masked and shaded in grey.**


The reconstruction skills of the REAL EST were compared to the UNI, RES, and VAR experiments, which defined the observation errors in different ways (Fig. 10 and Table. 2). The global mean skill scores for CC and CE were the best in the EST, whereas RES and VAR achieved comparable skills with the EST when the scaling parameter is carefully tuned. In contrast, UNI exhibits the lowest skill scores. In terms of RV, UNI performed remarkably than the others, when the scaling

factor or number of iterations was tuned with CC. Among the other experiments, EST maintained a relatively high RV, primarily because the observation errors in EST were smaller than those in RES and VAR (Fig. 10). Similar results were obtained when validating against GISTEMP (Lenssen et al., 2024; Fig. A2 and Table. A1).





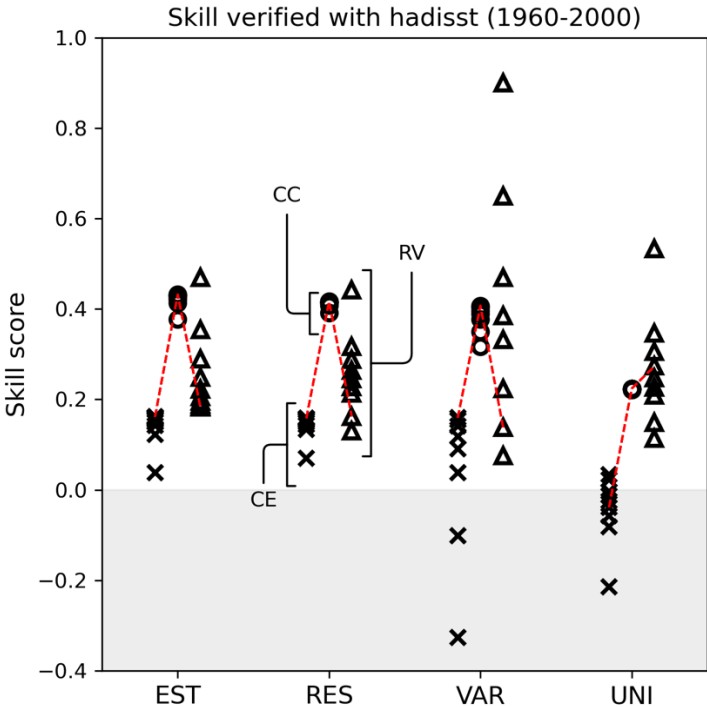

**Figure 10: Surface temperature reconstruction skill score for the REAL EST, RES, VAR, and UNI experiments based on the CC (circle), CE (triangle), and RV (square). Each mark corresponds to the skill score of a scaling factor or an iteration. The scores were calculated with HadISST for 1960-2000.**

Table 2: Global mean skill scores for REAL experiments verified with HadCRUT for 1960-2000

| Exp. Name | # of iteration / Scaling factor | CC | CE | RV |
|---|---|---|---|---|
| EST | 10 | **0.432** | **0.162** | 0.183 |
| UNI | 0.7 | 0.224 | -0.038 | **0.274** |
| VAR | 8 | 0.407 | 0.159 | 0.138 |
| RES | 1.5 | 0.416 | 0.158 | 0.162 |

Here we compared the observation errors in all the experiments except for UNI, since the skills based on CC and CE were exceptionally lower than those of the others. The observation errors in RES and VAR were roughly proportional to those in EST (Fig. 11a). This suggests that using a globally constant scaling factor may be reasonable for first-order approximation in



paleo-DA. However, a closer examination revealed large variations in the ratio of the errors in RES and VAR to those in EST. Specifically, the 10th and 90th percentile of the ratios were 1.84 and 4.37 for VAR and 1.79 and 5.29 for RES, respectively,

indicating significant spatial differences (Fig. 11b). This large variability also suggested that the observation errors cannot be optimally tuned using a globally constant factor. This, combined with the poor skills of UNI, underscores the importance of setting and tuning observation errors individually for each observation point, rather than applying a uniform universal adjustment.

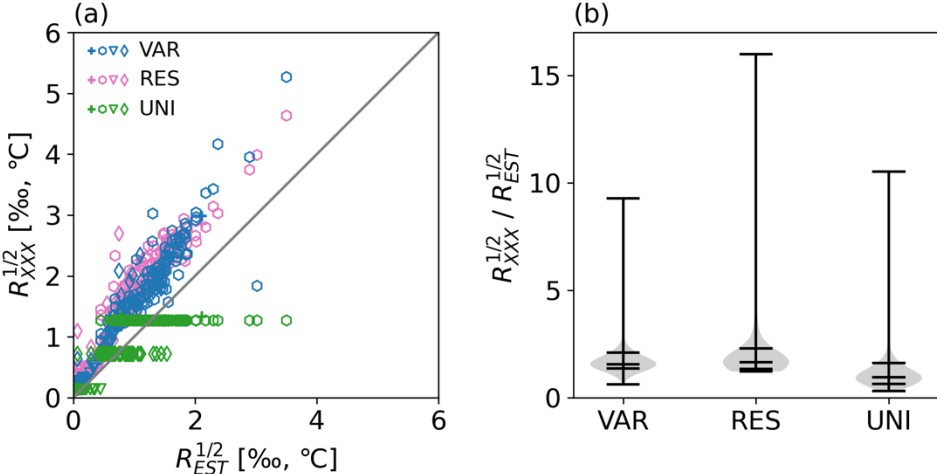


**Figure 11: (a) Comparison of observation errors in REAL EST and those in VAR (blue), RES (magenta), and UNI (green) in scatter plot. Cross indicates old documents; hexagon, ice cores; triangle, corals; and diamond, tree-ring cellulose. The observation errors are shown with physical units (°C or ‰). (b) Ratio of observation errors in REAL VAR, RES, and UNI to those in EST. Horizontal bars show the minimum, 10th, 50th, 90th, and maximum. $R_{XXX}^{1/2}$ denotes observations errors either in VAR, RES, or UNI.**


## 5 Limitations of the proposed method

### 5.1 Biases in background error covariance

This study used anomaly assimilation to mitigate model bias in the mean states and estimated the prior error covariance inflation factors to ensure that the variance of **B** used in the DA matched that of $\mathbf{B}_{est}$ (Eq. 19). This should ensure that biases

in the prior mean and the variance do not affect the reconstruction skills significantly. We investigated the impact of biases in the covariance structure among the model states, specifically the off-diagonal elements of **B**. In DA, the covariance plays a key role in spreading observation information spatially. If the prior covariance structure differs from the true covariance structure, incorrect increments are included in the prior update.





We investigated the impact of covariance bias using "BIAS", an experiment similar to the OSSE EST but with a different
model simulation for the background ensemble. Figure A3 shows the correlation between the mean surface temperature in the
NINO3 area and each model grid point and indicates that the nature run exhibited stronger correlations globally than the BIAS.

Figure 12 shows the estimated observation errors for the BIAS experiment and compares them with those from the OSSE EST
Rx16. The observation errors were consistently overestimated in BIAS for all the iterations. Although the iterative estimation
brings $\mathbf{R}_{est}$ closer to $\mathbf{R}_{tru}$, the discrepancy was larger than that in the unbiased case. Moreover, the ratio of $\mathbf{R}_{est}$ to the $\mathbf{R}_{tru}$
exhibited a wider distribution compared to that without bias, indicating greater uncertainty in error estimation. Nevertheless,
the reconstruction skills are all improved with $\mathbf{R}_{est}$, although the improvements were less pronounced than those without bias
(Fig. 13). These findings suggested that the innovation statistics remain effective even in the presence of model bias, and that
the reconstruction skills can still be improved through observation error estimations. The effectiveness of this method likely
depends on the magnitude of bias. However, the innovation statistics exhibited some tolerance to biases in $\mathbf{B}$, which is
consistent with previous findings (e.g., Li et al., 2009).

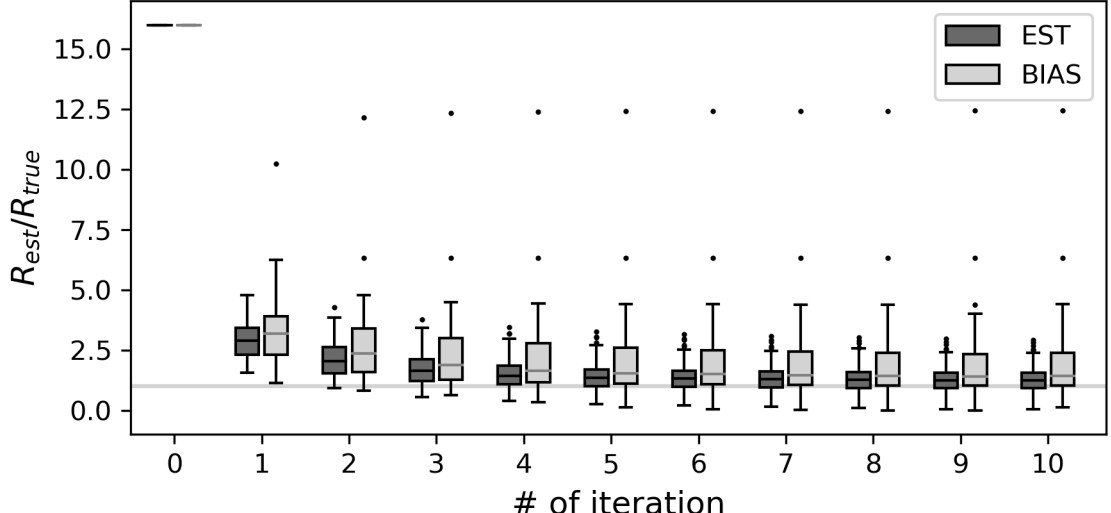

**Figure 12: Similar to Fig. 3 but for the OSSE EST (dark grey) and BIAS (grey). $\mathbf{R}_{ini}$ is Rx16 in both cases.**





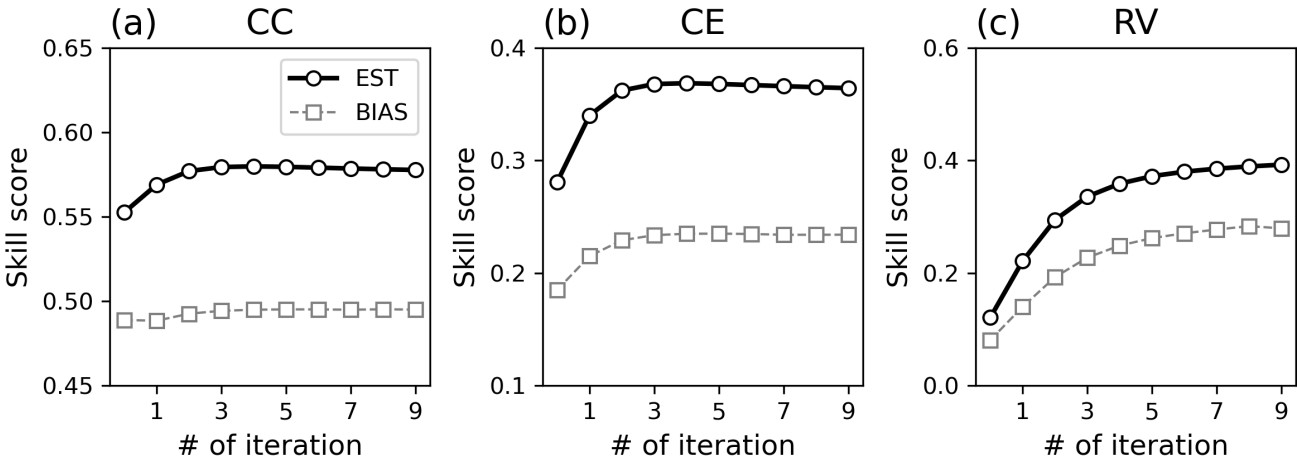

**Figure 13: Similar to Fig. 4 but for the OSSE EST (thick black line with circle) and BIAS (thin grey dashed line with square).**

### 5.2 Impact of sample size on observation error estimation

We investigated the impact of sample size on the accuracy of observation error estimations using a simplified two-variable model. The experimental design was the same as that of the OSSE EST, except for the background ensemble and observations:

1. First, background state $\mathbf{x}^b \in \mathbb{R}^2$ is generated by randomly sampling from a normal distribution with a mean of 0 and the variance of 1. Each element of $\mathbf{x}^b$ was designed to correlate with the other element with a correlation coefficient of 0.7. An ensemble of 136 members was generated in the same way.

2. Observations $\mathbf{y}^o \in \mathbb{R}^2$ are randomly sampled from a normal distribution with a mean of 0 and the variance of 1. The observations were generated for 131 time steps, assuming that the true state is always 0, meaning that the true observation error variance is 1.

3. Using the same background ensemble at every timesteps as in the stationary offline-DA, the analysis is computed by assimilating the observations. In the first iteration, the diagonal components of $\mathbf{R}$ are set to 2, which is twice the true observation variance.

4. Observation errors and covariance inflation factors are then estimated based on the analysis, prior, and observations using innovation statistics.

5. The 3rd and 4th steps are repeated, but with the estimated observation errors and the covariance inflation factors in DA. The set of analyses and estimations of the observation errors and inflation factors were repeated 20 times.

6. Steps 1 to 5 were repeated 100 times but with different realizations of $\mathbf{x}^b$ and $\mathbf{y}^o$.





Fig. 14 compares the estimated sizes of the observation errors after the $20^{th}$ iterations and $\langle \mathbf{d}_b^o(\mathbf{d}_b^o)^T \rangle$. Here, $\langle \mathbf{d}_b^o(\mathbf{d}_b^o)^T \rangle$ is expected to be 2 because the error variances of the prior and observation are both 1.0 (see Eq. 16). However, the estimated values varied between 1.2 to 2.7, highlighting the effect of sampling error. We observed a strong correlation between the estimated observation errors and $\langle \mathbf{d}_b^o(\mathbf{d}_b^o)^T \rangle$. Specifically, when $\langle \mathbf{d}_b^o(\mathbf{d}_b^o)^T \rangle$ is overestimated, the $\mathbf{R}_{est}$ is also overestimated, and vice versa. This occurs because $\mathbf{R}$ is estimated based on $\langle \mathbf{d}_b^o(\mathbf{d}_b^o)^T \rangle$ (see Eq. 18). Therefore, if $\langle \mathbf{d}_b^o(\mathbf{d}_b^o)^T \rangle$ is biased due to sampling noise, $\mathbf{R}_{est}$ will also be systematically biased.

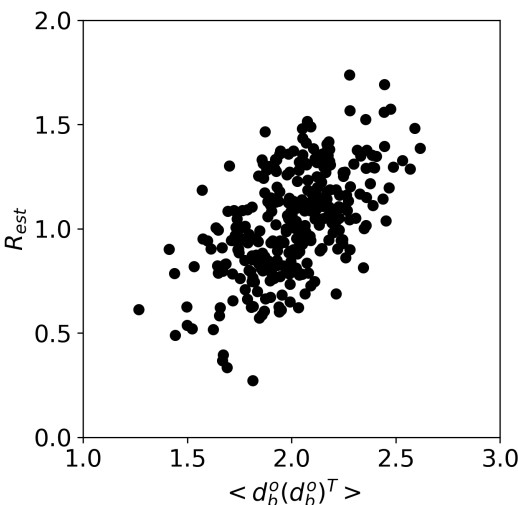

**Figure 14: Relationship between $\langle \mathbf{d}_b^o(\mathbf{d}_b^o)^T \rangle$ and $\mathbf{R}_{est}$ at the $20^{th}$ iterations for the simplified two-variable model.**

The $\mathbf{R}_{est}$ based on 131 samples was insufficient to suppress sampling noise even in a simplified two-variable model (Fig. 14), suggesting that the sample size used in Sects 3 and 4 (n=131) should also be too small for reliable observation error estimation. In case $\langle \mathbf{d}_b^o(\mathbf{d}_b^o)^T \rangle$ is biased, iterative estimation of observation errors can degrade the reconstruction skill score, because the estimation seeks to minimize the discrepancy between $\langle \mathbf{d}_b^o(\mathbf{d}_b^o)^T \rangle$ and $\mathbf{HBH}^T + \mathbf{R}$ used in DA. Thus, a larger sample size should be used for the more robust observation error estimation.

When the correlation between the model state variables is zero, $\langle \mathbf{d}_b^o(\mathbf{d}_b^o)^T \rangle$ explains almost all the variability in estimated observation errors (i.e., a correlation coefficient is nearly 1.0; not shown). However, as the correlation between the model states increased, the explained variance decreased (Fig. 14), likely because the observation error estimation at a given location



is influenced by $\langle \mathbf{d}_b^o (\mathbf{d}_b^o)^T \rangle$ in the surrounding points. When the estimation is affected by multiple surrounding observations, a simple linear relationship between the estimated error and $\langle \mathbf{d}_b^o (\mathbf{d}_b^o)^T \rangle$ does not hold true anymore.

## 6 Summary and conclusions

This study investigated the feasibility of observation error estimation using innovation statistics (Desroziers et al., 2005) within an offline-DA framework for paleoclimate reconstruction. We compared the performance of this approach in both an idealized framework assimilating pseudo-proxy data and a real case study assimilating actual proxy data. We found that the innovation statistics accurately estimated observation errors within the offline-DA scheme for the OSSE, achieving an absolute percent error of ~46%. Incorporating the estimated observation errors into DA improved reconstruction skill scores (CC and CE) by 5%-30%. Furthermore, we also found there are little dependence on the initial size of observation error variance used in DA in subsequent observation error estimation and analyses. Although the accuracy of the estimation depended on the sample size and the quality of the prior error covariance, the estimation method consistently improved the reconstruction skills.

Beyond idealized experiments, we demonstrated that the innovation statistics-based method also improved the reconstruction skills in real-world applications. The reconstruction skills with the estimated observation errors were comparable with or slightly better than those based on the variance of the observation or the differences between the observation and simulated observation-equivalent quantities. Although further tuning may result in better reconstruction skills in these other approaches, careful and manual parameter tuning is required, which is prohibitively time-consuming. In this regard, innovation statistics-based approach offers the advantage of automatically and systematically estimating errors.

DA nowadays is used to reconstruct climate of the ages deeper in the past such as the Eocene. In such applications, it is difficult to build linear regression models only with proxy data and instrument-based observations due to the shortage of overlapping periods, which is the commonly used approach in previous studies. As a result, observation errors based on the residual of the linear regression are not available. Not only for deep time paleo-DA but also for the late Holocene, direct assimilation of proxy data using process-based model is expected to be mainstream in the future as seen in the history of satellite data assimilation for NWP. The situation necessitates the development of other approaches to estimate observation errors. This study successfully demonstrated a feasible approach for paleoclimate reconstruction with DA. The method can be readily expanded to online-DA, since it was originally designed for this purpose. With more accurate observation errors, the observation impact estimates, such as analysis sensitivity to observation (e.g., Cardinali et al., 2004; Liu et al., 2009) and forecast sensitivity to observations in online-DA (e.g., Langland and Baker, 2004; Liu and Kalnay, 2008; Li et al., 2010) will be more accurate, too. These diagnostics can help to identify detrimental observations and/or key data sources for paleoclimate reconstruction. As such, the observation error estimation method should sophisticate and expand the possibility and accuracy of paleo-DA.

Despite the benefits, several challenges remain in the application of innovation statistics for offline-DA. Our study indicates that sampling noise may affect the accuracy of error estimation, especially with limited proxy records. If the sampling noise is not negligible, iterative estimation may worsen the reconstruction skill. To mitigate this, an iteration threshold should be set
to avoid any detrimental impact on the estimates. This issue was outside the scope of this study and requires future research. We did not consider age uncertainty in the proxy records. Although this is not vital for the present study or climate reconstruction in the last millennium, it is not true for deep-time paleo-DA. Age uncertainty can be considered as a misrepresentation in the archive model, a sub-model of the PSM (Evans et al., 2013). Accordingly, the uncertainty can be considered as a part of an observation error in DA. However, it remains unclear whether we should do so or not. Observation
errors that include age uncertainty can be much larger than prior errors. In such cases, the analysis can be reduced to the prior with little information from assimilated observations. To avoid this scenario, a method that separately accounts for age uncertainty is required. For the similar reason, the size of each error component must be evaluated, too.

In this study, we tested a specific method for estimating observation errors. However, several alternative approaches exist with
different complexities and applicability (e.g., Tandeo et al., 2020 and references therein). Other estimation methods should be explored for paleo-DA to refine observation error estimations.

Finally, it is important to emphasize that the estimated observation errors do not represent the true accuracy of the proxies in recording environmental conditions. Instead, as defined by Eq. 2, the estimated errors are specific to the DA system. Therefore,
the estimated observation errors are a system dependent, and not necessarily valid across systems. Consequently, the observation errors must be estimated separately for each reconstruction system.

**Appendices**

**A.1 Sensitivity to the observation error variance in R**

The sensitivity to the observation error variance was examined using the configuration of the OSSE. In this experiment, we
tested the effects of variation in $\mathbf{R} = k\,\mathbf{R}_{tru}$. The scaling factor was set to 0.25, 0.5, 1, 2, 4, 8, 16, or 32. The reconstruction skills ranged from 0.5 to 0.55 for CC and from 0.19 to 0.32 for CE, respectively, showing the importance of using accurate observation error $\mathbf{R}$ in DA.





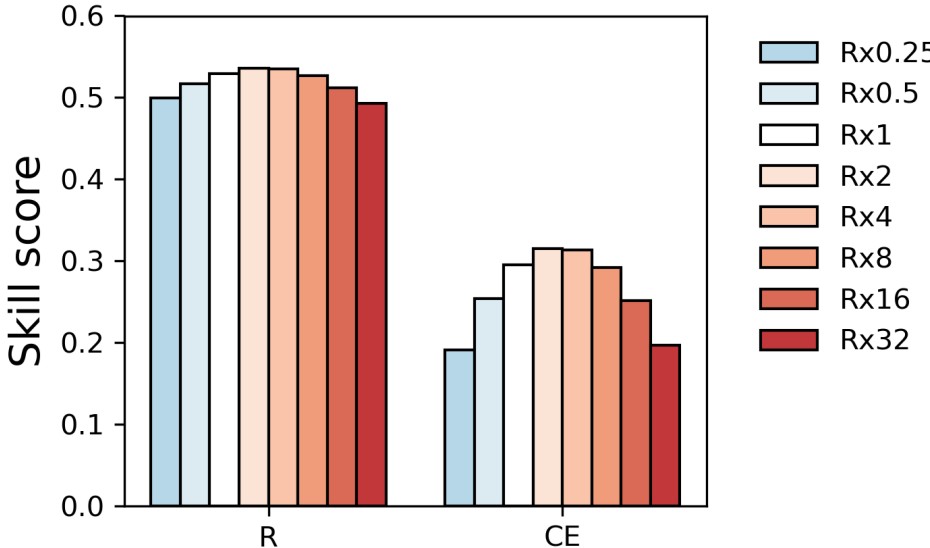

Figure A1: Sensitivity to the observation error variance in OSSE. The color of the bar indicates the scaling factor.

## A.2 Validation result with GISTEMP

The reconstruction skills of the REAL EST, UNI, RES, and VAR experiments were computed with GISTEMP (Lenssen et al., 2024; Fig. A2 and Table. A1). The similar tendency described in Sect. 4 was observed.



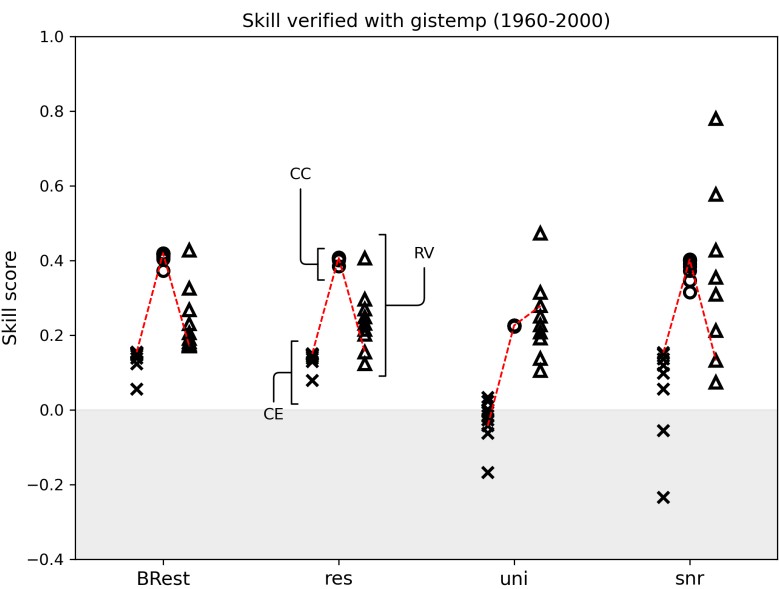

**Figure A2: Surface temperature reconstruction skill score for the REAL EST, RES, VAR, and UNI experiments based on the CC (circle), CE (triangle), and RV (square). The scores were calculated with GISTEMP for 1960-2000.**

**Table A1: Global mean skill scores for the REAL experiments verified with GISTEMP for 1960-2000**

| Exp. Name | # of iteration / Scaling factor | CC | CE | RV |
|---|---|---|---|---|
| EST | 10 | **0.420** | **0.155** | 0.170 |
| UNI | 0.6 | 0.226 | -0.042 | **0.278** |
| VAR | 8 | 0.403 | 0.153 | 0.133 |
| RES | 1.5 | 0.408 | 0.150 | 0.154 |

**A.3 Covariance structure of B in BIAS**

The BIAS experiment examined the impact of the biased off-diagonal term of **B**. The correlation between the mean surface temperature in the NINO3 area and that in each model grid point was mapped to show the covariance structure difference between the nature run and BIAS.



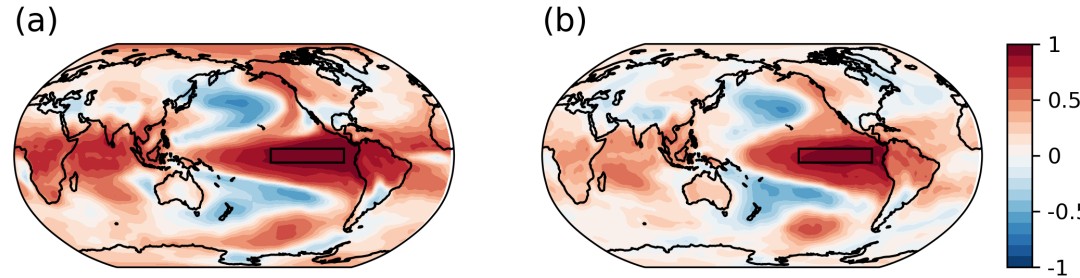


**Figure A3: Correlation coefficients between mean surface temperature in NINO3 area (rectangle area) and surface temperature in each model grid points for (a) nature run and (b) background ensemble used in the BIAS experiment.**

**Code availability**

The codes to reconstruct climate fields and estimate observation errors are written in Fortran and are available at Zenodo (https://zenodo.org/records/14987726)

**Data availability**

Proxy records used in this study were obtained from https://pastglobalchanges.org/science/wg/2k-network/Phase_3_Databases/Iso2k for Iso2k and https://pastglobalchanges.org/science/wg/2k-

network/Phase_2_Databases/Global_Temp/V2.0.0_2017 for PAGES2K. Gridded surface temperature data used to validate the reconstruction skills were obtained from https://www.metoffice.gov.uk/hadobs/hadcrut5/ for HadCRUT5 and https://data.giss.nasa.gov/gistemp/ for GISTEMP v4. CMIP5 MIROC5 historical runs which are used to drive the atmospheric GCM is available at https://esgf-node.llnl.gov/projects/cmip5/.

**Author contributions**

AO designed the study, conducted all the experiments, and drafted the manuscript. QD, JHW, and YK contributed to improve the study through the discussion about paleoclimate reconstruction. SK and DSC contributed to improve the study through the discussion about data assimilation and innovation statistics.

**Competing interests**

The authors declare that they have no conflict of interest.



**Acknowledgements**

This study was supported by Japan Society for the Promotion of Science KAKENHI Grants # 22H04938, # 22K14095, and
#24H00920.

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
