# Peer review of "Observation error estimation in climate proxies with data assimilation and innovation statistics"

_EGUsphere, 2025_

## Author Comment (AC1)

**Reply to Referee #1 (Prof. Lili Lei)**

Atsushi Okazaki, Diego S. Carrio, Quentin Dalaiden, Jarrah Harrison-Lofthouse, Shunji Kotsuki, Kei Yoshimura

We sincerely thank the reviewer for the positive assessment and the helpful comments and excellent suggestions. Below, we provide a point-to-point responses to all the reviewers' comments. The reviewers' comments are in *blue and italic*, and the replies are in black.

*Lines 195-200, it is hard to follow from (18) to (19). How the innovation statistics link to the covariance inflation? Using (17), is the numerator the same as the denominator, which give delta = 1? Moreover, in the following discussions, the role of the inflation, especially the relation with the observation error variance, is not clearly discussed.*

Thank you for the comment. The numerator in Eq. 19 represents the estimated size of trace($\mathbf{HBH}^\mathrm{T}$). In other words, it is the expected size of trace($\mathbf{HBH}^\mathrm{T}$). On the other hand, the denominator is the one represented by the ensemble. We can estimate how much the **B** (represented by the ensemble) should be inflated by calculating their ratio.

As for the second point, to accurately estimate the observation error, **B** should be estimated as well (Li et al., 2009). Depending on the magitude of the error variance **B**, the size of the estimated **R** differs. For instance, when **B** is underdispersive, the estimated **R** can be too small as well. To deal with this issue, we used the covariance inflation. We will update the sentences to make these points clearer in the revised manuscript.

*Lines 218-220, do you mean 136 annual mean simulations are used as ensemble priors? Are the simulations or anomalies used?*

Yes, 136 annual means (i.e., model states) are used as ensemble priors. While we use the raw simulated value for the state vector (**x**), the assimilation is performed using anomaly fields for the comparison between model states (**Hx**) and observation. We will modify the sentences for clarity in the revised manuscript.

*Lines 246-247, this is unclear. Do you mean the climatological mean is computed as a smoothing averaging with adjacent years? If yes, how many years are used to compute the climatological mean?*

Yes, exactly. We set a criteria of 30 year to calculate the climatological mean. If the overlapping period shorter than 30 years, the observation will not be assimilated. We will modify the sentences for clarity in the revised manuscript.

*Lines 269-275, till now, it is unclear why 'BIAS' is designed?*

Thank you for the comment. In general, the structure of the background error is considered to be different from the nature. We conducted the experiment 'BIAS' to investigate the impact of a misrepresented background error covariance.

*Lines 355-360, with too large (small) R, small (large) inflation values are expected. It would be nice to show the estimated inflation given different R.*

Thank you for the comment. The figure below shows the estimated inflation factors for (a) underestimated **R** and (b) overestimated **R**. As you expected, they are small with too large **R** and vice versa.

**(a) $R_{ini}$=0.25 x $R_{tru}$    (b) $R_{ini}$=16 x $R_{tru}$**

[Figure]

*Figure 9, please give some potential explanations for the regions with negative skills.*

We appreciate your valuable suggestion. If the estimation works well, the detrimental impact should be attributed to the bias in the prior covariance. We will add the discussion in the revised manuscript.

*Lines 414, 'remarkably' -> 'remarkably worse'?*

Thank you for finding the typo. We will correct it in the revised manuscript.

---

## Author Comment (AC2)

**Reply to Referee #2**

Atsushi Okazaki, Diego S. Carrio, Quentin Dalaiden, Jarrah Harrison-Lofthouse, Shunji Kotsuki, Kei Yoshimura

We sincerely thank the reviewer for the positive assessment and the helpful comments and excellent suggestions. Below, we provide a point-to-point responses to all the reviewers' comments. The reviewers' comments are in *blue and italic*, and the replies are in black.

*This paper provides an important contribution to our understanding of the uncertainty parameter (R) in paleoclimate applications of DA. As the authors note, this is an often-minor consideration of previous researchers, but nonetheless an important parameter. The innovation statistic application that is tested provides a useful alternative to the linear regression methods which requires significant overlap with 19th-21st century climate observations, but future tests will eventually be required to ensure that the proposed method is skillful for deep time reconstructions. The manuscript is well-structured and methodologically sound, and I recommend minor revisions prior to publication to further strengthen the clarity and accessibility of the work, as well as to understand impacts to the posterior ensemble.*

*Specifically, the text could be improved by additional plain language description of the innovation statistic method (see comment for line 126) and impact of age uncertainty (see comment for Line 554). Furthermore, an important role of R in paleoclimate studies is to properly quantify changes to the ensemble spread, and it would be beneficially to include additional analysis or commentary on how the innovation statistic impacts the ensemble range rather than just the mean.*

Thank you very much for your detailed review of our manuscript. The responses to the points raised here are written below.

*Line 126: This paragraph would benefit from expansion to provide a plain language summary of innovation statistics. The methods section is quite technical and will be difficult to follow for readers who are unfamiliar with the method.*

Thank you for the valuable suggestion. We will include a brief explanation of the innovation statistics in the revised manuscript.

*Line 144: Could you provide additional clarification on the difference between the LETKF and an EnKF implemented with a localization radius.*

Thank you for your comments. Simply put, they are a totally different idea. LETKF solves the update equation locally to enhance the computational efficiency, while the localization is a technique to mitigate the detrimental impact of the limited ensemble size.

*Line 203: Please explicitly state whether the OSSE is equivalent to a pseudoproxy experiment, or explain the differences if not.*

They are the same in the context of paleo-DA. We include this in the revised manuscript.

*Line 215: Does "MIROC5" refer to "MIROC5-iso" or a different simulation?*

They are different. "MIROC5-iso" is a name of the model we ran for this study. "MIROC5" referes to the model which participates in CMIP5 and provides data.

*Line 216: Why was only r1i1p1 used to create the prior? If this is the CMIP5 MIROC5 simulations, wouldn't more ensemble members be available?*

Yes, there are five ensemble runs in total. The purpose of using the two runs is to create an idealized situation, where the nature run and the prior model simulation are similar statistically. For that purpose, we used one for the nature run and the other for the prior. Therefore, we do not need more ensemble runs even though they are available.

*Line 221: Were proxies records filtered to span a certain amount of the 1870-2000 study period?*

Yes, the proxies and the model simulation have to overlap longer than 30 years. This treatment is necessary to calculate the anomaly of each. We will add the explanation to the revised manuscript.

*Line 224: What do you mean by "complementary"?*

The purpose of the paper is to estimate the observation errors of the climate proxies. The temperature data is instrumental data and not a climate proxy in this regard. This is the reason why we used the word. We will rephrase it in the revised manuscript.

*Line 224:  It's unclear why just the documentary data were used for temperature.*

Probably, there is a misunderstanding. We used surface temperature data recorded in the historical documents. The other proxies are also used for temperature reconstruction.

*Line 229: Could you describe the linear interpolation method. Is this an interpolation between two grid center points? How does this work in two-dimensional space?*

Thank you for raising this point. We calculated the weighted mean of the adjacent 4 grid points for the 2D interpolation. Specifically, we used bilinear interpolation. We will specify the interpolation method in the revised manuscript.

*Line 248: What metric(s) was optimized that resulted in a half-localization scale of 8,000 km?*

Thank you for pointing it out. We optimized the localization scale with the correlation. We will add the explanation to the revised manuscript.

*Line 295: Do these skill metrics consider the ensemble spread or just the ensemble mean?*

Thank you for the comment. We do not evaluate the ensemble spread in the manuscript. But, the discussion on RV is tightly connected with the ensemble spread. When RV is small, the observation error is large as discussed in Sect. 3. This means that DA weighs the simulation more, implying relatively small ensemble spread. In the revised manuscript, we will add more discussion about the ensemble spread.

*Line 295: Do these skill metrics consider the spatial correlation or interannual variability? If the former, how does the innovation statistic impact interannual variability in the posterior?*

Thank you for your comments. The metrics are calculated at all the model grid points, and then averaged spatially. Therefore, the correlation evaluates the interannual variability. We will add the explanation to the revised manuscript.

The innovation statistics change the observation error and consequently change the interannual variability. The reason is that DA combines a simulation and observations based on the corresponding errors. With larger observation errors, DA weighs the prior simulation more, and vice versa. The interannual variability of the analysis changes depending on the weights (errors), unless the correlation between the observation and the simulation is 1.0.

*Line 315: Please clarify the units within Figure 2.*

Thank you for pointing it out. The units are Celsius degree or permil.  The units will be added in the revised manuscript.

*Line 378: Is this because no PSM was applied?*

Thank you for raising the important point. It is possible that the SNR for ice cores are low because no PSM was applied to them. However, even with PSM, the estimated SNR should be smaller for ice cores because the skill of the PSM is still relatively low compared to the ones for coral and tree-ring cellulose (Okazaki and Yoshimura, 2019). We will add a discussion on that in the revised paper.

*Line 380: Also important to note that the small sample size of 3 records.*

We believe the small number of proxy points is not the cause of the small SNR. For each proxy point, more than 100 samples are used to estimate the SNR for each proxy point.

*Line 523: Please clarify what the 5%-30% improvement is measured against.*

They are measured against the reconstruction without observation error estimation.

*Line 554: Age uncertainty is a very important consideration for deep time. Not only is the exact date uncertain, but also the amount of time that each measurement represents, which will impact the variance and therefore the estimation of R.  Given the authors highlight deep-time applications as a key motivation, a more extensive discussion — or a small pilot analysis (e.g., assimilating non-annual records (i.e., speleothem) with age uncertainty) — would strengthen the case for broader applicability.*

We appreciate your important suggestion. Although we strongly agree with the idea that an additional analysis would enhance the value of the study, we would like to keep this as a future work because it is far beyond our scope; we need to develop another method to deal with the temporal uncertainty. Instead, we will expand the discussion in the revised manuscript.

---

## Author Response (AR1)

Dear Prof. Francesco Muschitiello,

Thank you for handling our manuscript "Observation error estimation in climate proxies with data assimilation and innovation statistics" and for your editorial guidance. We believe that the manuscript has been improved very much by considering the comments that two reviewers kindly gave to us.

Below, we provide point-to-point responses to all the reviewers' comments. The reviewers' comments are in *blue and italic*, the replies are in black, and the corresponding modifications are in light-gray box.

Best regards,

Atsushi Okazaki, Diego S. Carrio, Quentin Dalaiden, Jarrah Harrison-Lofthouse, Shunji Kotsuki, Kei Yoshimura

**For Reviewer #1 (Prof. Lili Lei)**

Lines 195-200, it is hard to follow from (18) to (19). How the innovation statistics link to the covariance inflation? Using (17), is the numerator the same as the denominator, which give delta = 1? Moreover, in the following discussions, the role of the inflation, especially the relation with the observation error variance, is not clearly discussed.

Thank you for the comment. The numerator in Eq. 19 represents the estimated size of trace(**HBH**T). In other words, it is the expected size of trace(**HBH**T). On the other hand, the denominator is the one represented by the ensemble. We can estimate how much the **B** (represented by the ensemble) should be inflated by calculating their ratio.

As for the second point, to accurately estimate the observation error **R**, the background error variance **B** should be estimated as well (Li et al., 2009). For instance, when **B** is underdispersive, **R** is esimated smaller. But this estimated **R** is not close to the true **R**. To deal with this issue, we used the covariance inflation. We updated the sentences to make these points clearer in the revised manuscript.

(L168-169) This study also estimates the covariance inflation factor because accurate **B** is required to estimate the observation error (Li et al., 2009).

(L196-200) The covariance inflation factor ( $\Delta$ ) can be estimated adaptively by comparing **HBH**T represented by the background ensemble and the estimated one with Eq. 17 (Li et al., 2009). This study estimated the factor for each model grid point (i.e., locally) following Miyoshi (2011).

$$\Delta = \frac{\mathit{trace} \left( \left\langle \mathbf{d}_b^a (\mathbf{d}_b^o)^T \right\rangle \circ \rho \circ \mathbf{R}^{-1} \right)}{\mathit{trace} \left( \mathbf{1} / (m-1) \, \mathbf{HX}^b (\mathbf{HX}^b)^T \circ \rho \circ \mathbf{R}^{-1} \right)}$$

Lines 218-220, do you mean 136 annual mean simulations are used as ensemble priors? Are the simulations or anomalies used?

Yes, 136 annual means (i.e., model states) are used as ensemble priors. While we use the raw simulated value for the state vector, the anomaly fields are used for the calculation of the innovation. We will modify the sentences for clarity in the revised manuscript.

As the averaging period used to compute the climatological mean varies across observations, this procedure is described in the section on **Data assimilation** rather than in the section on the **Background ensemble**.

(L223-225) We used a single-member simulation covering 1870-2005 to generate a 136-member background ensemble, where 136 annual means (i.e., model states) are used as an ensemble member (Steiger et al., 2014).

(L249-251) The assimilation was conducted following the anomaly-DA approach (e.g., Keenlyside et al., 2008; Smith et al., 2007), where the corresponding climatological mean is subtracted from both observations and background in the observation space to mitigate the detrimental impact of model bias.

Lines 246-247, this is unclear. Do you mean the climatological mean is computed as a smoothing averaging with adjacent years? If yes, how many years are used to compute the climatological mean?

Thank you very much for pointing this out. Such a treatment is possible, but in this study, the climatological means do not change over time. For the computation of the climatological mean, we set a criterion of 30 years. If the overlapping period is shorter than 30 years, the observation is discarded. We added the explanation in the revised manuscript.

(L251-254) We calculated the climatological mean using the overlapping years between observations and simulations during the period from 1900 to 2000. The overlapping period must span longer than 30 years for the computation of the climatological mean. Otherwise, the corresponding observation is discarded. Note that the period represented by the climatological mean differs by site since the observational period varies.

**Lines 269-275, till now, it is unclear why 'BIAS' is designed?**

Thank you for the comment. In general, the structure of the background error should be different from that of the nature. We conducted the experiment 'BIAS' to investigate the impact of a misrepresented background error covariance. We added the explanation in the revised manuscript.

(L265-267) In general, the structure of the background error is different from that of the nature. The experiment is designed to investigate the impact of a misrepresented background error covariance.

Lines 355-360, with too large (small) R, small (large) inflation values are expected. It would be nice to show the estimated inflation given different R.

Thank you for the comment. The figure below shows the estimated inflation factors for (a) underestimated **R** and (b) overestimated **R**. As you expected, they are small with too large **R** and vice versa. The figures and discussions were added in the revised manuscript.

(L337-341) The estimated inflation factors are shown in Fig. A2. Along with the iteration, the factors converged to a certain pattern after the 5th and 6th iterations as seen in  $\mathbf{R}_{\text{est}}$ . The inflation factors are globally smaller than 1.0 at each iteration. This suggests that the background ensemble should be overdispersive. The background ensemble includes the simulations for the late 20th century, which is strongly affected by the global warming. These states should not be reasonable ones for e.g., 19th century and caused the overdispersive background. This suggests the importance of selecting the background ensemble carefully.

(L384-390) With different  $\mathbf{R}_{\text{ini}}$ , the estimated inflation factors exhibit different spatial patterns at the first iteration, where large (small)  $\mathbf{R}_{\text{ini}}$  resulted in small (large) inflation factors (Fig. A2), as deduced from Eqs. 16, 17, and 18 and the fact that  $\langle \mathbf{d}_b^o(\mathbf{d}_b^o)^T \rangle$  is the same for all the experiments with different  $\mathbf{R}_{\text{ini}}$ . Nonetheless, the inflation factors converge to the similar patterns after the iterations (Fig. A2). The ensemble spread follows the inflation factors because otherwise the background ensemble is the same at all the analysis steps in this study (not shown).

The inflation factors and the ensemble spreads are tightly connected with  $\mathbf{R}_{\text{est}}$  and deducible by it. Thereby, we focus only on observation errors hereafter.

(L604-609)

**A.2 Estimated inflation factors**

The estimated inflation factors for the OSSE are shown in Fig. A2. The estimated inflation factors exhibit different spatial patterns at the first iteration with different Rini, where large (small) Rini resulted in small (large) inflation factors. Nonetheless, the inflation factors converge to the similar patterns after the iterations.

Figure A2: Estimated inflation factors for the OSSE (top) Rx16, (middle) Rx1, and (bottom) Rx0.25.

**Figure 9, please give some potential explanations for the regions with negative skills.**

We appreciate your valuable suggestion. If the estimation works well, the detrimental impact should be attributed to the bias in the prior covariance. We added the discussion in the revised manuscript.

(L422-423) In Siberia and south Pacific near the coast of south Chile, the skills decrease along with iteration, likely due to the bias in the background error covariance.

**Lines 414, 'remarkably' -> 'remarkably worse'?**

Thank you for finding the typo. Actually, the score of UNI is better than the others for RV. We corrected it in the revised manuscript.

(L438) UNI performed remarkably better than the others...

**For Reviewer #2**

This paper provides an important contribution to our understanding of the uncertainty parameter (R) in paleoclimate applications of DA. As the authors note, this is an often-minor consideration of previous researchers, but nonetheless an important parameter. The innovation statistic application that is tested provides a useful alternative to the linear regression methods which requires significant overlap with 19th-21st century climate observations, but future tests will eventually be required to ensure that the proposed method is skillful for deep time reconstructions. The manuscript is well-structured and methodologically sound, and I recommend minor revisions prior to publication to further strengthen the clarity and accessibility of the work, as well as to understand impacts to the posterior ensemble.

Specifically, the text could be improved by additional plain language description of the innovation statistic method (see comment for line 126) and impact of age uncertainty (see comment for Line 554). Furthermore, an important role of R in paleoclimate studies is to properly quantify changes to the ensemble spread, and it would be beneficially to include additional analysis or commentary on how the innovation statistic impacts the ensemble range rather than just the mean.

Thank you very much for your detailed review of our manuscript. We added a brief explanation of the innovation statistics and expanded the discussion on the age uncertainty. We also added discussion on the inflation factors as it can be regarded as an ensemble spread in the offline data assimilation.

Line 126: This paragraph would benefit from expansion to provide a plain language summary of innovation statistics. The methods section is quite technical and will be difficult to follow for readers who are unfamiliar with the method.

Thank you for the valuable suggestion. We included a brief explanation of the innovation statistics in the revised manuscript.

(L129-130) The statistics of innovations are called "innovation statistics" and contains the information on the observation and background errors.

Line 144: Could you provide additional clarification on the difference between the LETKF and an EnKF implemented with a localization radius.

Thank you for your comments. Simply put, they are a totally different idea. LETKF solves the update equation locally to enhance the computational efficiency, while the localization is a technique to mitigate the detrimental impact of the limited ensemble size.

Line 203: Please explicitly state whether the OSSE is equivalent to a pseudoproxy experiment, or explain the differences if not.

They are the same in the context of paleo-DA. We included it in the revised manuscript.

(L206-208) As such, OSSE can be considered as an idealized experiment and is equivalent to the idea of pseudoproxy experiment in paleo-DA (e.g., Steiger et al., 2014).

**Line 215: Does "MIROC5" refer to "MIROC5-iso" or a different simulation?**

They are different. "MIROC5-iso" is a name of the model we ran for this study. "MIROC5" referes to the model which participates in CMIP5 and provides data. For the clarity, we modified the sentence.

(L219-221) We derived the SST and SIC data from the Coupled Model Intercomparison Project Phase 5 (CMIP5; Taylor et al., 2012) historical simulation of MIROC5 with the "r1i1p1" ensemble member.

**Line 216: Why was only r1i1p1 used to create the prior? If this is the CMIP5 MIROC5 simulations, wouldn't more ensemble members be available?**

Yes, there are five ensemble members in total, and the ensemble size could be increased. It is generally true that a larger ensemble size improves the performance of ensemble-based data assimilation. However, we did not observe any apparent negative impact associated with the current ensemble size, and our conclusions remain the same even when the ensemble size is increased. Therefore, we consider the present setting to be adequate and have decided to retain it.

**Line 221: Were proxies records filtered to span a certain amount of the 1870-2000 study period?**

Yes, the proxies and the model simulation have to overlap longer than 30 years. This treatment is necessary to calculate the anomaly of each. We added the explanation to the revised manuscript.

(L252-253) The overlapping period must span longer than 30 years for the computation of the climatological mean. Otherwise, the corresponding observation is discarded.

**Line 224: What do you mean by "complementary"?**

The purpose of the paper is to estimate the observation errors of the climate proxies. The temperature data is instrumental data and not a climate proxy in this regard. This is the reason why we used the word. We rephrased it in the revised manuscript.

(L229-230) In addition, three surface temperature records in historical documents from the PAGES2k database are used for the ease of comparison of the observation errors with the previous studies.

**Line 224: It's unclear why just the documentary data were used for temperature.**

Probably, there is a misunderstanding. We used surface temperature data recorded in the historical documents. The other proxies are also used for temperature reconstruction. We modified the sentence for the clarification.

(L229-230) In addition, three surface temperature records in historical documents from the PAGES2k database are used for the ease of comparison of the observation errors with the previous studies.

**Line 229: Could you describe the linear interpolation method. Is this an interpolation between two grid center points? How does this work in two-dimensional space?**

Thank you for raising this point. We calculated the weighted mean of the adjacent 4 grid points for the 2D interpolation. Specifically, we used bilinear interpolation. We specified the interpolation method in the revised manuscript.

(L234) The model fields at the proxy locations were extracted using bilinear interpolation...

**Line 248: What metric(s) was optimized that resulted in a half-localization scale of 8,000 km?**

Thank you for pointing it out. We optimized the localization scale with the correlation. We added the explanation to the revised manuscript.

(L255-257) The localization scale was manually tuned beforehand to maximize the correlation coefficient, and a half-localization scale of 8,000 km was used for all the experiments.

**Line 295: Do these skill metrics consider the ensemble spread or just the ensemble mean?**

Thank you for the comment. We do not evaluate the ensemble spread in the manuscript. But, the discussion on inflation factor and RV is tightly connected with the ensemble spread. When RV is small, the observation error is large as discussed in Sect. 3. This means that DA weighs the simulation more, implying relatively small ensemble spread. In the revised manuscript, we added more discussions about the inflation factor and RV.

(L337-341) The estimated inflation factors are shown in Fig. A2. Along with the iteration, the factors converged to a certain pattern after the 5th and 6th iterations as seen in  $\mathbf{R}_{\text{est}}$ . The inflation factors are globally smaller than 1.0 at each iteration. This suggests that the background ensemble should be overdispersive. The background ensemble includes the simulations for the late 20th century, which is strongly affected by the global warming. These states should not be reasonable ones for e.g., 19th century and caused the overdispersive background. This suggests the importance of selecting the background ensemble carefully.

(L384-390) With different  $\mathbf{R}_{\text{ini}}$ , the estimated inflation factors exhibit different spatial patterns at the first iteration, where large (small)  $\mathbf{R}_{\text{ini}}$  resulted in small (large) inflation factors (Fig. A2), as deduced from Eqs. 16, 17, and 18 and the fact that  $\langle \mathbf{d}_b^o(\mathbf{d}_b^o)^T \rangle$  is the same for all the experiments with different  $\mathbf{R}_{\text{ini}}$ . Nonetheless, the inflation factors converge to the similar patterns after the iterations (Fig. A2). The ensemble spread follows the inflation factors because otherwise the background ensemble is the same at all the analysis steps in this study (not shown).

The inflation factors and the ensemble spreads are tightly connected with  $\mathbf{R}_{\text{est}}$  and deducible by it. Thereby, we focus only on observation errors hereafter.

(L604-609)

**A.2 Estimated inflation factors**

The estimated inflation factors for the OSSE are shown in Fig. A2. The estimated inflation factors exhibit different spatial patterns at the first iteration with different Rini, where large (small) Rini resulted in small (large) inflation factors. Nonetheless, the inflation factors converge to the similar patterns after the iterations.

Figure A2: Estimated inflation factors for the OSSE (top) Rx16, (middle) Rx1, and (bottom) Rx0.25.

Line 295: Do these skill metrics consider the spatial correlation or interannual variability? If the former, how does the innovation statistic impact interannual variability in the posterior?

Thank you for your comments. The metrics are calculated at each model grid point, and then averaged spatially. Therefore, the correlation evaluates the interannual variability. We added the explanation to the revised manuscript.

The innovation statistics change the observation error and consequently change the interannual variability. The reason is that DA combines a simulation and observations based on the corresponding errors. With larger observation errors, DA weighs the prior simulation more, and vice versa. The interannual variability of the analysis changes depending on the weights (errors), unless the temporal correlation between the observation and the simulation is 1.

(L308-309) Here,  $x_i^a$  and  $x_i^{ref}$  denote i th year of the analysis and the reference. Accordingly, the metrics evaluates the interannual variability.

**Line 315: Please clarify the units within Figure 2.**

Thank you for pointing it out. The units are Celsius degree or permil. The units were added in the revised manuscript.

(L324-325) Figure 2: Relationship between true and estimated observation error variances at each iteration step for the OSSE with initial observation variance of Rx16. The units of the axes are all % or  $^{\circ}$ C.

**Line 378: Is this because no PSM was applied?**

Thank you for raising the important point. It is possible that the SNR for ice cores are low because no PSM was applied to them. However, even with PSM, the estimated SNR should be smaller for ice cores because the skill of the PSM is still relatively low compared to the ones for coral and tree-ring cellulose (Okazaki and Yoshimura, 2019). We added a discussion on that in the revised paper.

(L400-403) This may be because no PSM was applied for ice cores. However, even with PSM, the estimated SNR should be smaller for ice cores because the skill of the PSM is found to be relatively low compared to the ones for coral and tree-ring cellulose (Okazaki and Yoshimura, 2019).

**Line 380: Also important to note that the small sample size of 3 records.**

We believe the small number of proxy points is not the cause of the small SNR. For each proxy point, more than 100 samples are used to estimate the SNR for each proxy point. Here, the number of sample is the number of data in one proxy record (=number of years).

**Line 523: Please clarify what the 5%-30% improvement is measured against.**

They are measured against the reconstruction without observation error estimation.

(L546-547) Incorporating the estimated observation errors into DA improved reconstruction skill scores (CC and CE) by 5%-30% compared to the ones without them.

Line 554: Age uncertainty is a very important consideration for deep time. Not only is the exact date uncertain, but also the amount of time that each measurement represents, which will impact the variance and therefore the estimation of R. Given the authors highlight deep-time applications as a key motivation, a more extensive discussion — or a small pilot analysis (e.g., assimilating non-annual records (i.e., speleothem) with age uncertainty) — would strengthen the case for broader applicability.

We appreciate your important suggestion. We strongly agree that an additional analysis on the age uncertainty enhance the value of the study. However, we would like to keep this as a future

work because it is far beyond our scope; we need to develop another method to deal with the temporal uncertainty. Instead, we expanded the discussion in the revised manuscript.

(L577-585) We did not consider age uncertainty on the exact date and the length of the representative period of the proxy records. Although this is not vital for the present study or climate reconstruction in the last millennium, it is not true for deep-time paleo-DA. Even for the last millennium, it may not be negligible when aiming to reconstruct climate at a monthly or finer temporal resolution. Age uncertainty can be considered as a misrepresentation in the archive model, a sub-model of the PSM (Evans et al., 2013). Accordingly, the uncertainty can be regarded as a part of an observation error in DA. However, it remains unclear whether we should do so or not. Observation errors that include age uncertainty can be much larger than prior errors. In such cases, the analysis is reduced to the prior with little information from assimilated observations. To avoid this scenario, a method that separately accounts for age uncertainty (e.g., Osman et al., 2021) and/or a refinement of the dating (e.g., Furukawa et al., 2017) is required. For the similar reason, the size of each error component must be evaluated, too.